# FROM STARS TO SUBGRAPHS: UPLIFTING ANY GNN WITH LOCAL STRUCTURE AWARENESS

**Lingxiao Zhao**
Carnegie Mellon Uni.
lingxiao@cmu.edu

**Wei Jin**
Michigan State Uni.
jinwei2@msu.edu

**Leman Akoglu**
Carnegie Mellon Uni.
lakoglu@andrew.cmu.edu

**Neil Shah**
Snap Inc.
nshah@snap.com

## ABSTRACT

Message Passing Neural Networks (MPNNs) are a common type of Graph Neural Network (GNN), in which each node's representation is computed recursively by aggregating representations ("messages") from its immediate neighbors akin to a star-shaped pattern. MPNNs are appealing for being efficient and scalable, however their expressiveness is upper-bounded by the 1st-order Weisfeiler-Leman isomorphism test (1-WL). In response, prior works propose highly expressive models at the cost of scalability and sometimes generalization performance. Our work stands between these two regimes: we introduce a general framework to uplift *any* MPNN to be more expressive, with limited scalability overhead and greatly improved practical performance. We achieve this by extending local aggregation in MPNNs from star patterns to general subgraph patterns (e.g., $k$-egonets): in our framework, each node representation is computed as the encoding of a surrounding induced subgraph rather than encoding of immediate neighbors only (i.e. a star). We choose the subgraph encoder to be a GNN (mainly MPNNs, considering scalability) to design a general framework that serves as a wrapper to uplift any GNN. We call our proposed method GNN-AK (GNN As Kernel), as the framework resembles a convolutional neural network by replacing the kernel with GNNs. Theoretically, we show that our framework is strictly more powerful than 1&2-WL, and is not less powerful than 3-WL. We also design subgraph sampling strategies which greatly reduce memory footprint and improve speed while maintaining performance. Our method sets new state-of-the-art performance by large margins for several well-known graph ML tasks; specifically, 0.08 MAE on ZINC, 74.79% and 86.887% accuracy on CIFAR10 and PATTERN respectively.

## 1 INTRODUCTION

Graphs are permutation invariant, combinatorial structures used to represent relational data, with wide applications ranging from drug discovery, social network analysis, image analysis to bioinformatics (Duvenaud et al., 2015; Fan et al., 2019; Shi et al., 2019; Wu et al., 2020). In recent years, Graph Neural Networks (GNNs) have rapidly surpassed traditional methods like heuristically defined features and graph kernels to become the dominant approach for graph ML tasks.

Message Passing Neural Networks (MPNNs) (Gilmer et al., 2017) are the most common type of GNNs owing to their intuitiveness, effectiveness and efficiency. They follow a recursive aggregation mechanism where each node aggregates information from its immediate neighbors repeatedly. However, unlike simple multi-layer feedforward networks (MLPs) which are universal approximators of continuous functions (Hornik et al., 1989), MPNNs cannot approximate all permutation-invariant graph functions (Maron et al., 2019b). In fact, their expressiveness is upper bounded by the first order Weisfeiler-Leman (1-WL) isomorphism test (Xu et al., 2018). Importantly, researchers have shown that such 1-WL equivalent GNNs are not expressive, or powerful, enough to capture basic structural concepts, i.e., counting motifs such as cycles or triangles (Zhengdao et al., 2020; Arvind et al., 2020) that are shown to be informative for bio- and chemo-informatics (Elton et al., 2019).

The weakness of MPNNs urges researchers to design more expressive GNNs, which are able to discriminate graphs from an isomorphism test perspective; Chen et al. (2019) prove the equivalence between such tests and universal permutation invariant function approximation, which theoretically justifies it. As $k$-WL is strictly more expressive than 1-WL, many works (Morris et al., 2019; 2020b) try to incorporate $k$-WL in the design of more powerful GNNs, while others approach $k$-WL

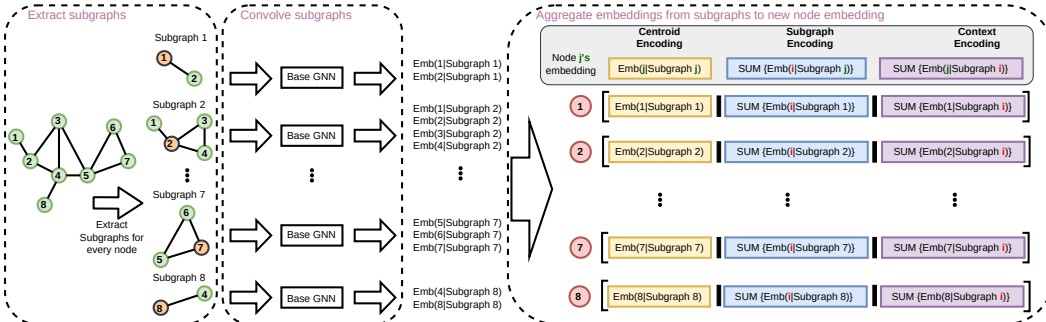

Figure 1: **Shown**: one GNN-AK$^+$ layer. For each layer, GNN-AK$^+$ first extracts $n$ (# nodes) rooted subgraphs, and convolves all subgraphs with a base GNN as kernel, producing multiple rich subgraph-node embeddings of the form $\mathsf{Emb}(i \mid \mathrm{Sub}[j])$ (node $i$'s embedding when applying a GNN kernel on subgraph $j$). From these, we extract and concatenate three encodings for a given node $j$: (i) centroid $\mathsf{Emb}(j \mid \mathrm{Sub}[j])$, (ii) subgraph $\sum_i \mathsf{Emb}(i \mid \mathrm{Sub}[j])$, and (iii) context $\sum_i \mathsf{Emb}(j \mid \mathrm{Sub}[i])$. GNN-AK$^+$ repeats the process for $L$ layers, then sums all resulting node embeddings to compute the final graph embedding. As a weaker version, GNN-AK only contains encodings (i) and (ii).

expressiveness indirectly from matrix invariant operations (Maron et al., 2019a;b; Keriven & Peyré, 2019) and matrix language perspectives (Balcilar et al., 2021). However, they require O($k$)-order tensors to achieve $k$-WL expressiveness, and thus are not scalable or feasible for application on large, practical graphs. Besides, the bias-variance tradeoff between complexity and generalization (Neal et al., 2018) and the fact that almost all graphs (i.e. O($2^{\binom{n}{2}}$) graphs on $n$ vertices, Babai et al. (1980)) can be distinguished by 1-WL challenge the necessity of developing such extremely expressive models. In a complementary line of work, Loukas (2020a) sheds light on developing more powerful GNNs while maintaining linear scalability, finding that MPNNs can be universal approximators provided that nodes are sufficiently distinguishable. Relatedly, several works propose to add features to make nodes more distinguishable, such as identifiers (Loukas, 2020a), subgraph counts (Bouritsas et al., 2020), distance encoding (Li et al., 2020), and random features (Sato et al., 2021; Abboud et al., 2021). However, these methods either focus on handcrafted features which lose the premise of automatic learning, or create permutation sensitive features that hurt generalization.

**Present Work.** Our work stands between the two regimes of extremely expressive but unscalable $k$-order GNNs, and the limited expressiveness yet high scalability of MPNNs. Specifically, we propose a general framework that serves as a "wrapper" to uplift **any** GNN. We observe that MPNNs' local neighbor aggregation follows a star pattern, where the representation of a node is characterized by applying an injective aggregator function as an encoder to the star subgraph (comprised of the central node and edges to neighbors). We propose a design which naturally generalizes from encoding the star to encoding a more flexibly defined subgraph, and we replace the standard injective aggregator with a GNN: in short, we characterize the new representation of a node by using a GNN to encode a locally induced encompassing subgraph, as shown in Fig.1. This uplifts GNN as a base model in effect by applying it on each *subgraph* instead of the whole input graph. This generalization is close to Convolutional Neural Networks (CNN) in computer vision: like the CNN that convolves image patches with a kernel to compute new pixel embeddings, our designed wrapper convolves subgraphs with a GNN to generate new node embeddings. Hence, we name our approach GNN-AK (GNN As Kernel). We show theoretically that GNN-AK is strictly more powerful than 1&2-WL with any MPNN as base model, and is not less powerful than 3-WL with PPGN (Maron et al., 2019a) used. We also give sufficient conditions under which GNN-AK can successfully distinguish two non-isomorphic graphs. Given this increase in expressive power, we discuss careful implementation strategies for GNN-AK, which allow us to carefully leverage multiple modalities of information from subgraph encoding, and resulting in an empirically more expressive version GNN-AK$^+$. As a result, GNN-AK and GNN-AK$^+$ induce a constant factor overhead in memory. To amplify our method's practicality, we further develop a subgraph sampling strategy inspired by Dropout (Srivastava et al., 2014) to drastically reduce this overhead (1-3$\times$ in practice) without hurting performance. We conduct extensive experiments on 4 simulation datasets and 5 well-known real-world graph classification & regression benchmarks (Dwivedi et al., 2020; Hu et al., 2020), to show significant and consistent practical benefits of our approach across different MPNNs and datasets. Specifically, GNN-AK$^+$ sets new state-of-the-art performance on ZINC, CIFAR10, and PATTERN – for example, on ZINC we see a relative error reduction of 60.3%, 50.5%, and 39.4% for base model being

GCN (Kipf & Welling, 2017), GIN (Xu et al., 2018), and (a variant of) PNA (Corso et al., 2020) respectively. To summarize, our contributions are listed as follows:

- **A General GNN-AK Framework.** We propose GNN-AK (and enhanced GNN-AK$^+$), a general framework which uplifts any GNN by encoding local subgraph structure with a GNN.
- **Theoretical Findings.** We show that GNN-AK's expressiveness is strictly better than 1&2-WL, and is not less powerful than 3-WL. We analyze sufficient conditions for successful discrimination.
- **Effective and Efficient Realization.** We present effective implementations for GNN-AK and GNN-AK$^+$ to fully exploit all node embeddings within a subgraph. We design efficient online subgraph sampling to mitigate memory and runtime overhead while maintaining performance.
- **Experimental Results.** We show strong empirical results, demonstrating both expressivity improvements as well as practical performance gains where we achieve new state-of-the-art performance on several graph-level benchmarks.

Our implementation is easy-to-use, and directly accepts any GNN from PyG (Fey & Lenssen, 2019) for plug-and-play use. See code at `https://github.com/GNNAsKernel/GNNAsKernel`.

## 2 RELATED WORK

Exploiting subgraph information in GNNs is not new; in fact, $k$-WL considers all $k$ node subgraphs. Monti et al. (2018); Lee et al. (2019) exploit motif information within aggregation, and others (Bouritsas et al., 2020; Barceló et al., 2021) augment MPNN features with handcrafted subgraph based features. MixHop (Abu-El-Haija et al., 2019) directly aggregates $k$-hop information by using adjacency matrix powers, ignoring neighbor connections. Towards a meta-learning goal, G-meta (Huang & Zitnik, 2020) applies GNNs on rooted subgraphs around each node to help transferring ability. Tahmasebi & Jegelka (2020) only theoretically justifies subgraph convolution with GNN by showing its ability in counting substructures. Zhengdao et al. (2020) also represent a node by encoding its local subgraph, however using non-scalable relational pooling. $k$-hop GNN (Nikolentzos et al., 2020) uses $k$-egonet in a specially designed way: it encodes a rooted subgraph via sequentially passing messages from $k$-th hops in the subgraph to $k - 1$ hops, until it reaches the root node, and use the root node as encoding of the subgraph. Ego-GNNs (Sandfelder et al., 2021) computes a context encoding with SGC (Wu et al., 2019) as the subgraph encoder, and only be studied on node-level tasks. Both $k$-hop GNN and Ego-GNNs can be viewed as a special case of GNN-AK. You et al. (2021) designs ID-GNNs which inject node identity during message passing with the help of $k$-egonet, with $k$ being the number of layers of GNN (Hamilton et al., 2017). Unlike GNN-AK which uses rooted subgraphs, Fey et al. (2020); Thiede et al. (2021); Bodnar et al. (2021a) design GNNs to use certain subgraph patterns (like cycles and paths) in message passing, however their preprocessing requires solving the subgraph isomorphism problem. Cotta et al. (2021) explores reconstructing a graph from its subgraphs. A contemporary work (Zhang & Li, 2021) also encodes rooted subgraphs with a base GNN but it essentially views a graph as a bag of subgraphs while GNN-AK modifies the 1-WL color refinement and has many iterations. Viewing graph as a bag of subgraphs is also explored in another contemporary work (Bevilacqua et al., 2022). To summarize, our work differs by (i) proposing a general subgraph encoding framework motivated from theoretical Subgraph-1-WL for uplifting GNNs, and (ii) addressing scalability issues involved with using subgraphs, which poses significant challenges for subgraph-based methods in practice. **See additional related work in Appendix.A.1.**

## 3 GENERAL FRAMEWORK AND THEORY

We first introduce our setting and formalisms. Let $G = (\mathcal{V}, \mathcal{E})$ be a graph with node features $\mathbf{x}_i \in \mathbb{R}^d$, $\forall i \in \mathcal{V}$. We consider graph-level problems where the goal is to classify/regress a target $y_G$ by learning a graph-level representation $\mathbf{h}_G$. Let $\mathcal{N}_k(v)$ be the set of nodes in the $k$-hop egonet rooted at node $v$. $\mathcal{N}(v) = \mathcal{N}_1(v) \backslash v$ denotes the immediate neighbors of node $v$. For $\mathcal{S} \subseteq \mathcal{V}$, let $G[\mathcal{S}]$ be the induced subgraph: $G[\mathcal{S}] = (\mathcal{S}, \{(i, j) \in \mathcal{E} | i \in \mathcal{S}, j \in \mathcal{S}\})$. Then $G[\mathcal{N}_k(v)]$ denotes the $k$-hop egonet rooted at node $v$. We also define $Star(v) = (\mathcal{N}_1(v), \{(v, j) \in \mathcal{E} | j \in \mathcal{N}(v)\})$ be the induced star-like subgraph around $v$. We use $\{\cdot\}$ denotes multiset, i.e. set that allows repetition.

Before presenting GNN-AK, we highlight the insights in designing GNN-AK and driving the expressiveness boost. **Insight 1: Generalizing star to subgraph.** In MPNNs, every node aggregates information from its immediate neighbors following a star pattern. Consequently, MPNNs fail to distinguish any non-isomorphic regular graphs where all stars are the same, since all nodes have the same degree. Even simply generalizing star to the induced, 1-hop egonet considers connec-

tions among neighbors, enabling distinguishing regular graphs. **Insight 2: Divide and conquer.** When two graphs are non-isomorphic, there exists a subgraph where this difference is captured (see Figure 3). Although a fixed-expressiveness GNN may not distinguish the two original graphs, it may distinguish the two *smaller* subgraphs, given that the required expressiveness for successful discrimination is proportional to graph size (Loukas, 2020b). As such, GNN-AK divides the harder problem of encoding the whole graph to smaller and easier problems of encoding its subgraphs, and "conquers" the encoding with the base GNN.

### 3.1 FROM STARS TO SUBGRAPHS

We first take a close look at MPNNs, identifying their limitations and expressiveness bottleneck. MPNNs repeatedly update each node's embedding by aggregating embeddings from their neighbors a fixed number of times (layers) and computing a graph-level embedding $\mathbf{h}_G$ by global pooling. Let $\mathbf{h}_v^{(l)}$ denote the $l$-th layer embedding of node $v$. Then, MPNNs compute $\mathbf{h}_G$ by

$$\mathbf{h}_v^{(l+1)} = \phi^{(l)}\left(\mathbf{h}_v^{(l)}, f^{(l)}\left(\{\mathbf{h}_u^{(l)}|u \in \mathcal{N}(v)\}\right)\right) \quad l = 0, ..., L-1 \; ; \quad \mathbf{h}_G = \text{POOL}(\{\mathbf{h}_v^{(L)}|v \in \mathcal{V}\}) \quad (1)$$

where $\mathbf{h}_i^{(0)} = \mathbf{x}_i$ is the original features, $L$ is the number of layers, and $\phi^{(l)}$ and $f^{(l)}$ are the $l$-th layer update and aggregation functions. $\phi^{(l)}$, $f^{(l)}$ and POOL vary among different MPNNs and influence their expressiveness and performance. MPNNs achieve maximum expressiveness (1-WL) when all three functions are injective (Xu et al., 2018).

MPNNs' expressiveness upper bound follows from its close relation to the 1-WL isomorphism test (Morris et al., 2019). Similar to MPNNs which repeatedly aggregate self and neighbor representations, at $t$-th iteration, for each node $v$, 1-WL test aggregates the node's own label (or color) $c_v^{(t)}$ and its neighbors' labels $\{c_u^{(t)}|u \in \mathcal{N}(v)\}$, and hashes this multi-set of labels $\left\{c_v^{(t)}, \{c_u^{(t)}|u \in \mathcal{N}(v)\}\right\}$ into a new, compressed label $c_v^{(t+1)}$. 1-WL outputs the set of all node labels $\left\{c_v^{(T)}|v \in \mathcal{V}\right\}$ as $G$'s fingerprint, and decides two graphs to be non-isomorphic as soon as their fingerprints differ.

The hash process in 1-WL outputs a new label $c_v^{(t+1)}$ that uniquely characterizes the star graph $Star(v)$ around $v$, i.e. two nodes $u, v$ are assigned different compressed labels only if $Star(u)$ and $Star(v)$ differ. Hence, it is easy to see that when two non-isomorphic unlabeled (i.e., all nodes have the same label) $d$-regular graphs have the same number of nodes, 1-WL cannot distinguish them. This failure limits the expressiveness of 1-WL, but also identifies its bottleneck: the star is not distinguishing enough. Instead, we propose to generalize the star $Star(v)$ to subgraphs, such as the egonet $G[\mathcal{N}_1(v)]$ and more generally $k$-hop egonet $G[\mathcal{N}_k(v)]$. This results in an improved version of 1-WL which we call *Subgraph-1-WL*. Formally,

**Definition 3.1** (**Subgraph-1-WL**). Subgraph-1-WL generalizes the 1-WL graph isomorphism test algorithm by replacing color refinement (at iteration $t$) by $c_v^{(t+1)} = \text{HASH}(Star^{(t)}(v))$ with $c_v^{(t+1)} = \text{HASH}(G^{(t)}[\mathcal{N}_k(v)]), \forall v \in \mathcal{V}$ where $\text{HASH}(\cdot)$ is an injective function on graphs.

Note that an injective hash function for star graphs is equivalent to that for multi-sets, which is easy to derive (Zaheer et al., 2017). In contrast, Subgraph-1-WL must hash a general subgraph , where an injective hash function for graphs is non-trivial (as hard as graph isomorphism). Thus, we derive a variant called Subgraph-1-WL* by using a weaker choice for $\text{HASH}(\cdot)$ – specifically, 1-WL. Effectively, we *nest* 1-WL inside Subgraph-1-WL. Formally,

**Definition 3.2** (**Subgraph-1-WL***). Subgraph-1-WL* is a less expressive variant of Subgraph-1-WL where $c_v^{(t+1)} = \text{1-WL}(G^{(t)}[\mathcal{N}_k(v)])$.

We further transfer Subgraph-1-WL to neural networks, resulting in GNN-AK whose expressiveness is upper bounded by Subgraph-1-WL. The natural transformation with maximum expressiveness is to replace the hash function with a universal subgraph encoder of $G[\mathcal{N}_k(v)]$, which is non-trivial as it implies solving the challenging graph isomorphism problem in the worst case. Analogous to using 1-WL as a weaker choice for $\text{HASH}(\cdot)$ inside Subgraph-1-WL*, we can use use any GNN (most practically, MPNN) as an encoder for subgraph $G[\mathcal{N}_k(v)]$. Let $G^{(l)}[\mathcal{N}_k(v)] = G[\mathcal{N}_k(v)|\mathbf{H}^{(l)}]$ be the attributed subgraph with hidden features $\mathbf{H}^{(l)}$ at the $l$-th layer. Then, GNN-AK computes $\mathbf{h}_G$ by

$$\mathbf{h}_v^{(l+1)} = \text{GNN}^{(l)}\left(G^{(l)}[\mathcal{N}_k(v)]\right) \quad l = 0, ..., L-1 \quad ; \quad \mathbf{h}_G = \text{POOL}(\{\mathbf{h}_v^{(L)}|v \in \mathcal{V}\}) \quad (2)$$

Notice that GNN-AK acts as a "wrapper" for any base GNN (mainly MPNN). This uplifts its expressiveness as well as practical performance as we demonstrate in the following sections.

## 3.2 THEORY: EXPRESSIVENESS ANALYSIS

We next theoretically study the expressiveness of GNN-AK, by investigating the expressiveness of Subgraph-1-WL. We first establish that GNN-AK and Subgraph-1-WL have the same expressiveness under certain conditions. A GNN is able to distinguish two graphs if its embeddings for two graphs are not identical. A GNN is said to have the same expressiveness as a graph isomorphism test when for any two graphs the GNN outputs different embeddings if and only if (iff) the isomorphism test deems them non-isomorphic. We give following theorems with proof in Appendix.

**Theorem 1.** *When the base model is an MPNN with sufficient number of layers and injective $\phi$, $f$ and POOL functions shown in Eq. (1), and MPNN-AK has an injective POOL function shown in Eq. (2), then MPNN-AK is as powerful as Subgraph-1-WL$^*$.*

See Appendix.A.3 for proof. A more general version of Theorem 1 is that GNN-AK is as powerful as Subgraph-1-WL iff base GNN of GNN-AK is as powerful as the HASH function of Subgraph-1-WL in distinguishing subgraphs, following the same proof logic. The Theorem implies that we can characterize expressiveness of GNN-AK through studying Subgraph-1-WL and Subgraph-1-WL$^*$.

**Theorem 2.** *Subgraph-1-WL$^*$ is strictly more powerful than 1&2-WL[1].*

A direct corollary from Theorem 1&2 is as follows, which is empirically verified in Table 1.

**Corollary 2.1.** *When MPNN is 1-WL expressive, MPNN-AK is strictly more powerful than 1&2-WL.*

**Theorem 3.** *When HASH($\cdot$) is 3-WL expressive, Subgraph-1-WL is no less powerful than 3-WL, that is, it can discriminate some graphs for which 3-WL fails.*

A direct corollary from Theorem 1&3 is as follows, which is empirically verified in Table 1.

**Corollary 3.1.** *PPGN-AK can distinguish some 3-WL-failed non-isomorphic graphs.*

**Theorem 4.** *For any $k \geq 3$, there exists a pair of $k$-WL-failed graphs that **cannot** be distinguished by Subgraph-1-WL even with injective HASH($\cdot$) when $t$-hop egonets are used with $t \leq 4$.*

Theorem 4 is proven (Appendix.A.6) by observing that with limited $t$ all rooted subgraphs of two non-isomorphic graphs from $CFI(k)$ family (Cai et al., 1992) are isomorphic, i.e. local rooted subgraph is not enough to capture the "global" difference. This opens a future direction of generalizing rooted subgraph to general subgraph (as in $k$-WL) while keeping number of subgraphs in O($|\mathcal{V}|$).

**Proposition 1** (sufficient conditions). *For two non-isomorphic graphs $G, H$, Subgraph-1-WL with $k$-egonet can successfully distinguish them if: 1) for any node reordering $v_1^G, ..., v_{|\mathcal{V}_G|}^G$ and $v_1^H, ..., v_{|\mathcal{V}_H|}^H$, $\exists i \in [1, \max(|\mathcal{V}_G|, |\mathcal{V}_H|)]$ that $G[\mathcal{N}_k(v_i^G)]$ and $H[\mathcal{N}_k(v_i^H)]$ are non-isomorphic[2]; and 2) HASH($\cdot$) is discriminative enough that HASH($G[\mathcal{N}_k(v_i^G)]$) $\neq$ HASH($H[\mathcal{N}_k(v_i^H)]$).*

This implies subgraph size should be large enough to capture difference, but not larger which requires more expressive base model (Loukas, 2020a). We empirically verify Prop. 1 in Table 2.

## 4 CONCRETE REALIZATION

We first realize GNN-AK with two type of encodings, and then present an empirically more expressive version, GNN-AK$^+$, with (i) an additional *context encoding*, and (ii) a *subgraph pooling* design to incorporate distance-to-centroid, readily computed during subgraph extraction. Next, we discuss a random-walk based rooted subgraph extraction for graphs with small diameter to reduce memory footprint of $k$-hop egonets. We conclude this section with time and space complexity analysis.

**Notation.** Let $G = (\mathcal{V}, \mathcal{E})$ be the graph with $N = |\mathcal{V}|$, $G^{(l)}[\mathcal{N}_k(v)]$ be the $k$-hop egonet rooted at node $v \in \mathcal{V}$ in which $\mathbf{h}_u^{(l)}$ denotes node $u$'s hidden representation for $u \in \mathcal{N}_k(v)$ at the $l$-th layer of GNN-AK. To simplify notation, we use $\mathrm{Sub}^{(l)}[v]$ instead of $G^{(l)}[\mathcal{N}_k(v)]$ to indicate the the attribute-enriched induced subgraph for $v$. We consider all intermediate node embeddings across rooted subgraphs. Specifically, let $\mathsf{Emb}(i \mid \mathrm{Sub}^{(l)}[j])$ denote node $i$'s embedding when applying base GNN$^{(l)}$ on $\mathrm{Sub}^{(l)}[j]$; we consider node embeddings for every $j \in \mathcal{V}$ and every $i \in \mathrm{Sub}^{(l)}[j]$. Note

---

[1]1-WL and 2-WL are known to be equally powerful, see Azizian & Lelarge (2021) and Maron et al. (2019a).
[2]When $|\mathcal{V}_G| < |\mathcal{V}_H|$, $\forall i \in \{|\mathcal{V}_G|, |\mathcal{V}_G| + 1, ..., |\mathcal{V}_H|\}$, let $G[\mathcal{N}_k(v_i^G)]$ denote an empty subgraph.

that the base GNN can have multiple convolutional layers, and Emb refers to the node embeddings at the last layer before global pooling $\text{POOL}_{\text{GNN}}$ that generates subgraph-level encoding.

**Realization of GNN-AK.** We can formally rewrite Eq. (2) as

$$\mathbf{h}_v^{(l+1)|\text{Subgraph}} = \text{GNN}^{(l)}\Big(\text{Sub}^{(l)}[v]\Big) := \text{POOL}_{\text{GNN}^{(l)}}\Big(\big\{\text{Emb}(i \mid \text{Sub}^{(l)}[v]) \mid i \in \mathcal{N}_k(v)\big\}\Big) \quad (3)$$

We refer to the encoding of the rooted subgraph $\text{Sub}^{(l)}[v]$ in Eq. (3) as the *subgraph encoding*. Typical choices of $\text{POOL}_{\text{GNN}^{(l)}}$ are SUM and MEAN. As each rooted subgraph has a root node, $\text{POOL}_{\text{GNN}^{(l)}}$ can additionally be realized to differentiate the root node by self-concatenating its own representation, resulting in the following realization as each layer of GNN-AK:

$$\mathbf{h}_v^{(l+1)} = \text{FUSE}\Big(\mathbf{h}_v^{(l+1)|\text{Centroid}}, \ \mathbf{h}_v^{(l+1)|\text{Subgraph}}\Big) \quad \text{where } \mathbf{h}_v^{(l+1)|\text{Centroid}} := \text{Emb}(v \mid \text{Sub}^{(l+1)}[v]) \quad (4)$$

where FUSE is concatenation or sum, and $\mathbf{h}_v^{(l)|\text{Centroid}}$ is referred to as the *centroid encoding*. The realization of GNN-AK in Eq.4 closely follows the theory in Sec.3.

**Realization of GNN-AK$^+$.** We further develop GNN-AK$^+$, which is more expressive than GNN-AK, based on two observations. **First**, we observe that Eq.4 does not fully exploit all information inside the rich intermediate embeddings generated for Eq.4, and propose an additional *context encoding*.

$$\mathbf{h}_v^{(l+1)|\text{Context}} := \text{POOL}_{\text{Context}}\Big(\big\{\text{Emb}(v \mid \text{Sub}^{(l)}[j]) \mid \forall j \text{ s.t. } v \in \mathcal{N}_k(j)\big\}\Big) \quad (5)$$

Different from subgraph and centroid encodings, the context encoding captures views of node $v$ from different subgraph contexts, or points-of-view. **Second**, GNN-AK extracts the rooted subgraph for every node with efficient $k$-hop propagation (complexity $O(k|\mathcal{E}|)$), along which the distance-to-centroid (D2C)[3] within each subgraph is readily recorded at no additional cost and can be used to augment node features; (Li et al., 2020) shows this theoretically improves expressiveness. Therefore, we propose to uses the D2C by default in two ways in GNN-AK$^+$: (i) augmenting hidden representation $\mathbf{h}_v^{(l)}$ by concatenating it with the encoding of D2C; (ii) using it to gate the subgraph and context encodings before $\text{POOL}_{\text{Subgraph}}$ and $\text{POOL}_{\text{Context}}$, with the intuition that embeddings of nodes far from $v$ contribute differently from those close to $v$.

To formalize the gate mechanism guided by D2C, let $\mathbf{d}_{i|j}^{(l)}$ be the encoding of distance from node $i$ to $j$ at $l$-th layer[4]. Applying gating changes Eq. (5) to

$$\mathbf{h}_{\text{gated},v}^{(l+1)|\text{Context}} := \text{POOL}_{\text{Context}}\Big(\big\{\text{Sigmoid}(\mathbf{d}_{v|j}^{(l)}) \odot \text{Emb}(v \mid \text{Sub}^{(l)}[j]) \mid \forall j \text{ s.t. } v \in \mathcal{N}_k(j)\big\}\Big) \quad (6)$$

where $\odot$ denotes element-wise multiplication. Similar changes apply to Eq. (3) to get $\mathbf{h}_{\text{gated},v}^{(l)|\text{Subgraph}}$.

Formally, each layer of GNN-AK$^+$ is defined as

$$\mathbf{h}_v^{(l+1)} = \text{FUSE}\Big(\mathbf{d}_{i|j}^{(l+1)}, \ \mathbf{h}_v^{(l+1)|\text{Centroid}}, \ \mathbf{h}_{\text{gated},v}^{(l+1)|\text{Subgraph}}, \ \mathbf{h}_{\text{gated},v}^{(l+1)|\text{Context}}\Big) \quad (7)$$

where FUSE is concatenation or sum. We illustrate in Figure 1 the $l$-th layer of GNN-AK$^{(+)}$.

**Proposition 2.** *GNN-AK$^+$ is at least as powerful as GNN-AK.* See Proof in Appendix.A.A.8.

**Beyond $k$-egonet Subgraphs.** The $k$-hop egonet (or $k$-egonet) is a natural choice in our framework, but can be too large when the input graph's diameter is small, as in social networks (Kleinberg, 2000), or when the graph is dense. To limit subgraph size, we also design a random-walk based subgraph extractor. Specifically, to extract a subgraph rooted at node $v$, we perform a fixed-length random walk starting at $v$, resulting in visited nodes $\mathcal{N}_{rw}(v)$ and their induced subgraph $G[\mathcal{N}_{rw}(v)]$. In practice, we use adaptive random walks as in Grover & Leskovec (2016). To reduce randomness, we use multiple truncated random walks and union the visited nodes as $\mathcal{N}_{rw}(v)$. Moreover, we employ online subgraph extraction during training that re-extracts subgraphs at every epoch, which further alleviates the effect of randomness via regularization.

**Complexity Analysis.** Assuming $k$-egonets as rooted subgraphs, and an MPNN as base model. For each graph $G = (\mathcal{V}, \mathcal{E})$, the subgraph extraction takes $O(k|\mathcal{E}|)$ runtime complexity, and outputs $|\mathcal{V}|$ subgraphs, which collectively can be represented as a union graph $\mathcal{G}_\cup = (\mathcal{V}_\cup, \mathcal{E}_\cup)$ with $|\mathcal{V}|$ disconnected components, where $|\mathcal{V}_\cup| = \sum_{v \in \mathcal{V}} |\mathcal{N}_k(v)|$ and $|\mathcal{E}_\cup| = \sum_{v \in \mathcal{V}} |\mathcal{E}_{G[\mathcal{N}_k(v)]}|$. GNN-AK$^{(+)}$ can be viewed as applying base GNN on the union graph. Assuming base GNN has $O(|\mathcal{V}| + |\mathcal{E}|)$ runtime and memory complexity, GNN-AK$^{(+)}$ has $O(|\mathcal{V}_\cup| + |\mathcal{E}_\cup|)$ runtime and memory cost. For rooted subgraphs of size $s$, GNN-AK$^{(+)}$ induces an $O(s)$ factor overhead over the base model.

---

[3]We record D2C value for every node in every subgraph, and the value is categorical instead of continuous.

[4]The categorical D2C does not change across layers, but is encoded with different parameters in each layer.

## 5 Improving Scalability: SubgraphDrop

The complexity analysis reveals that GNN-AK($^+$) introduce a constant factor overhead (subgraph size) in runtime and memory over the base GNN. Subgraph size can be naturally reduced by choosing smaller $k$ for $k$-egonet, or by ranking and truncating visited nodes in a random walk setting. However limiting to very small subgraphs tends to hurt performance as we empirically show in Table 6. Here, we present a different subsampling-based approach that carefully selects only a subset of the $|\mathcal{V}|$ rooted subgraphs. Further more, inspired from Dropout (Srivastava et al., 2014), we only drop subgraphs during training while still use all subgraphs when evaluation. Novel strategies are designed specifically for three type of encodings to eliminate the estimation bias between training and evaluation. We name it SubgraphDrop for dropping subgraphs during training. SubgraphDrop significantly reduces memory overhead while keeping performance nearly the same as training with all subgraphs. We first present subgraph sampling strategies, then introduce the designs of aligning training and evaluation. Fig. 2 in Appendix.A.2 provides a pictorial illustration.

### 5.1 Subgraph Sampling Strategies

Intuitively, if $u, v$ are directly connected in $G$, subgraphs $G[\mathcal{N}_k(u)]$ and $G[\mathcal{N}_k(v)]$ share a large overlap and may contain redundancy. With this intuition, we aim to sample only $m \ll |\mathcal{V}|$ minimally redundant subgraphs to reduce memory overhead. We propose three fast sampling strategies (See Appendix.A.2) that select subgraphs to evenly cover the whole graph, where each node is covered by $\approx R$ (redundancy factor) selected subgraphs; $R$ is a hyperparameter used as a sampling stopping condition. Then, GNN-AK($^+$)-S (with Sampling) has roughly $R$ times the overhead of base model ($R \leq 3$ in practice). We remark that our subsampling strategies are randomized and fast, which are both desired characteristics for an online Dropout-like sampling in training.

### 5.2 Training with SubgraphDrop

Although dropping redundant subgraphs greatly reduces overhead, it still loses information. Thus, as in Dropout (Srivastava et al., 2014), we only "drop" subgraphs during training while still using all of them during evaluation. Randomness in sampling strategies enforces that selected subgraphs differ across training epochs, preserving most information due to amortization. On the other hand, it makes it difficult for the three encodings during training to align with full-mode evaluation. Next, we propose an alignment procedure for each type of encoding.

**Subgraph and Centroid Encoding in Training.** Let $\mathcal{S} \subseteq \mathcal{V}$ be the set of root nodes of the selected subgraphs. When sampling during training, subgraph and centroid encoding can only be computed for nodes $v \in \mathcal{S}$, following Eq. (3) and Eq. (4), resulting incomplete subgraph encodings $\{\mathbf{h}_v^{(l)|\text{Subgraph}} | v \in \mathcal{S}\}$ and centroid encodings $\{\mathbf{h}_v^{(l)|\text{Centroid}} | v \in \mathcal{S}\}$. To estimate uncomputed encodings of $u \in \mathcal{V} \setminus \mathcal{S}$, we propose to *propagate* encodings from $\mathcal{S}$ to $\mathcal{V} \setminus \mathcal{S}$. Formally, let $k_{max} = \max_{u \in \mathcal{V} \setminus \mathcal{S}} \text{Dist}(u, \mathcal{S})$ where $\text{Dist}(u, \mathcal{S}) = \min_{v \in \mathcal{S}} \text{ShortestPathDistance}(u, v)$. Then we partition $\mathcal{U} = \mathcal{V} \setminus \mathcal{S}$ into $\{\mathcal{U}_1, ..., \mathcal{U}_{k_{max}}\}$ with $\mathcal{U}_d = \{u \in \mathcal{U} | \text{Dist}(u, \mathcal{S}) = d\}$. We propose to spread vector encodings of $\mathcal{S}$ out iteratively, i.e. compute vectors of $\mathcal{U}_d$ from $\mathcal{U}_{d-1}$. Formally, we have

$$\mathbf{h}_u^{(l)|\text{Subgraph}} = \text{Mean}\big(\{\mathbf{h}_v^{(l)|\text{Subgraph}} | v \in \mathcal{U}_{d-1}, (u, v) \in \mathcal{E}\}\big) \quad \text{for} \quad d = 1, 2 \ldots k_{max}, \forall u \in \mathcal{U}_d, \quad (8)$$

$$\mathbf{h}_u^{(l)|\text{Centroid}} = \text{Mean}\big(\{\mathbf{h}_v^{(l)|\text{Centroid}} | v \in \mathcal{U}_{d-1}, (u, v) \in \mathcal{E}\}\big) \quad \text{for} \quad d = 1, 2 \ldots k_{max}, \forall u \in \mathcal{U}_d, \quad (9)$$

**Context Encoding in Training.** Following Eq. (5), context encodings can be computed for every node $v \in \mathcal{V}$ as each node is covered at least $R$ times during training with SubgraphDrop. However when $\text{POOL}_{\text{Context}}$ is $\text{SUM}(\cdot)$, the scale of $\mathbf{h}_v^{(l)|\text{Context}}$ is smaller than the one in full-mode evaluation. Thus, we scale the context encodings up to align with full-mode evaluation. Formally,

$$\mathbf{h}_v^{(l)|\text{Context}} = \frac{|\{j \in \mathcal{V} | \mathcal{N}_k(j) \ni v\}|}{|\{j \in \mathcal{S} | \mathcal{N}_k(j) \ni v\}|} \times \text{SUM}\Big(\{\text{Emb}(v \mid \text{Sub}^{(l)}[j]) \mid \forall j \in \mathcal{S} \text{ s.t. } v \in \mathcal{N}_k(j)\}\Big) \quad (10)$$

When $\text{POOL}_{\text{Context}}$ is $\text{MEAN}(\cdot)$, the context encoding is computed without any modification.

## 6 Experiments

In this section we (1) empirically verify the expressiveness benefit of GNN-AK($^+$) on 4 simulation datasets; (2) show GNN-AK($^+$) boosts practical performance significantly on 5 real-world datasets; (3) demonstrate the effectiveness of SubgraphDrop; (4) conduct ablation studies of concrete designs. We mainly report the performance of GNN-AK$^+$, while still keep the performance of GNN-AK with GIN as base model for reference, as it is fully explained by our theory.

**Simulation Datasets:** 1) EXP (Abboud et al., 2021) contains 600 pairs of 1&2-WL failed graphs that are splited into two classes where each graph of a pair is assigned to two different classes. 2) SR25 (Balcilar et al., 2021) has 15 strongly regular graphs (3-WL failed) with 25 nodes each. SR25 is translated to a 15 way classification problem with the goal of mapping each graph into a different class. 3) Substructure counting (i.e. triangle, tailed triangle, star and 4-cycle) problems on random graph dataset (Zhengdao et al., 2020). 4) Graph property regression (i.e. connectedness, diameter, radius) tasks on random graph dataset (Corso et al., 2020). All simulation datasets are used to empirically verify the expressiveness of GNN-AK($^+$). **Large Real-world Datasets:** ZINC-12K, CIFAR10, PATTER from Benchmarking GNNs (Dwivedi et al., 2020) and MolHIV, and MolPCBA from Open Graph Benchmark (Hu et al., 2020). **Small Real-world Datasets**: MUTAG, PTC, PRO-TEINS, NCI1, IMDB, and REDDIT from TUDatset (Morris et al., 2020a) (their results are presented in Appendix A.12). See Table 5 in Appendix for all dataset statistics.

**Baselines.** We use GCN (Kipf & Welling, 2017), GIN (Xu et al., 2018), PNA$^*$ [5] (Corso et al., 2020), and 3-WL powerful PPGN (Maron et al., 2019a) directly, which also server as base model of GNN-AK to see its general uplift effect. GatedGCN (Dwivedi et al., 2020), DGN (Beani et al., 2021), PNA (Corso et al., 2020), GSN(Bouritsas et al., 2020), HIMP (Fey et al., 2020), and CIN (Bodnar et al., 2021a) are referenced directly from literature for real-world datasets comparison. Hyperparameter and model configuration are described in Appendix.A.10.

## 6.1 EMPIRICAL VERIFICATION OF EXPRESSIVENESS

Table 1: Simulation dataset performance: *GNN-AK($^+$) boosts base GNN across tasks, empirically verifying expressiveness lift.* (ACC: accuracy, MAE: mean absolute error, OOM: out of memory)

| Method | EXP (ACC) | SR25 (ACC) | Counting Substructures (MAE) | | | | Graph Properties ($\log_{10}$(MAE)) | | |
|---|---|---|---|---|---|---|---|---|---|
| | | | Triangle | Tailed Tri. | Star | 4-Cycle | IsConnected | Diameter | Radius |
| GCN | 50% | 6.67% | 0.4186 | 0.3248 | 0.1798 | 0.2822 | -1.7057 | -2.4705 | -3.9316 |
| GCN-AK$^+$ | **100%** | 6.67% | 0.0137 | 0.0134 | 0.0174 | 0.0183 | -2.6705 | -3.9102 | -5.1136 |
| GIN | 50% | 6.67% | 0.3569 | 0.2373 | 0.0224 | 0.2185 | -1.9239 | -3.3079 | -4.7584 |
| GIN-AK | **100%** | 6.67% | 0.0934 | 0.0751 | 0.0168 | 0.0726 | -1.9934 | -3.7573 | -5.0100 |
| GIN-AK$^+$ | **100%** | 6.67% | 0.0123 | 0.0112 | 0.0150 | 0.0126 | -2.7513 | -3.9687 | -5.1846 |
| PNA$^*$ | 50% | 6.67% | 0.3532 | 0.2648 | 0.1278 | 0.2430 | -1.9395 | -3.4382 | -4.9470 |
| PNA$^*$-AK$^+$ | **100%** | 6.67% | 0.0118 | 0.0138 | 0.0166 | 0.0132 | -2.6189 | -3.9011 | -5.2026 |
| PPGN | 100% | 6.67% | 0.0089 | 0.0096 | 0.0148 | 0.0090 | -1.9804 | -3.6147 | -5.0878 |
| PPGN-AK$^+$ | 100% | **100%** | OOM | OOM | OOM | OOM | OOM | OOM | OOM |

Table 1 presents the results on simulation datasets. To save space we present GNN-AK$^+$ with different base models but only one one version of GNN-AK: GIN-AK. All GNN-AK($^+$) variants perform perfectly on EXP, while only PPGN alone do so previously. Moreover, PPGN-AK$^+$ reaches perfect accuracy on SR25, while PPGN fails. Similarly, GNN-AK($^+$) consistently boosts all MPNNs for substructure and graph property prediction (PPGN-AK$^+$ is OOM as it is quadratic in input size).

Table 2: PPGN-AK expressiveness on SR25.

| Base PPGN's #L | 1-hop egonet | | | 2-hop egonet | | |
|---|---|---|---|---|---|---|
| PPGN-AK's #L | 1 | 2 | 3 | 1 | 2 | 3 |
| 1 | 26.67% | 100% | 100% | 26.67% | 26.67% | 46.67% |
| 2 | 33.33% | 100% | 100% | 26.67% | 26.67% | 53.33% |
| 3 | 26.67% | 100% | 100% | 33.33% | 26.67% | 53.33% |

In Table 2 we look into PPGN-AK's performance on SR25 as a function of $k$-egonets ($k \in [1, 2]$), as well as the number of PPGN (inner) layers and (outer) iterations for PPGN-AK. We find that at least 2 inner layers is needed with 1-egonet subgraphs to achieve top performance. With 2-egonets more inner layers helps, although performance is sub-par, attributed to PPGN's disability to distinguish larger (sub)graphs, aligning with Proposition 1 (Sec. 3.2).

## 6.2 COMPARING WITH SOTA AND GENERALITY

Having studied expressiveness tasks, we turn to performance on real-world datasets, as shown in Table 3. We observe similar performance lifts across all datasets and base GNNs (we omit PPGN due to scalability), demonstrating our framework's generality. Remarkably, GNN-AK$^+$ sets new SOTA performance for several benchmarks – ZINC, CIFAR10, and PATTERN – with a relative error reduction of 60.3%, 50.5%, and 39.4% for base model being GCN, GIN, and PNA$^*$ respectively.

---

[5]PNA$^*$ is a variant of PNA that changes from using degree to scale embeddings to encoding degree and concatenate to node embeddings. This eliminates the need of computing average degree of datasets in PNA.

Table 3: Real-world dataset performance: *GNN-AK$^+$ achieves SOTA performance for ZINC-12K, CIFAR10, and PATTERN.* (OOM: out of memory, −: missing values from literature)

| Method | ZINC-12K (MAE) | CIFAR10 (ACC) | PATTERN (ACC) | MolHIV (ROC) | MolPCBA (AP) |
|---|---|---|---|---|---|
| GatedGCN | $0.363 \pm 0.009$ | $69.37 \pm 0.48$ | $84.480 \pm 0.122$ | — | — |
| HIMP | $0.151 \pm 0.006$ | — | — | $0.7880 \pm 0.0082$ | — |
| PNA | $0.188 \pm 0.004$ | $70.47 \pm 0.72$ | $86.567 \pm 0.075$ | $0.7905 \pm 0.0132$ | $0.2838 \pm 0.0035$ |
| DGN | $0.168 \pm 0.003$ | $72.84 \pm 0.42$ | $86.680 \pm 0.034$ | $0.7970 \pm 0.0097$ | $0.2885 \pm 0.0030$ |
| GSN | $0.115 \pm 0.012$ | — | — | $0.7799 \pm 0.0100$ | — |
| CIN | $\mathbf{0.079 \pm 0.006}$ | — | — | $\mathbf{0.8094 \pm 0.0057}$ | — |
| GCN | $0.321 \pm 0.009$ | $58.39 \pm 0.73$ | $85.602 \pm 0.046$ | $0.7422 \pm 0.0175$ | $0.2385 \pm 0.0019$ |
| GCN-AK$^+$ | $0.127 \pm 0.004$ | $72.70 \pm 0.29$ | $\mathbf{86.887 \pm 0.009}$ | $0.7928 \pm 0.0101$ | $0.2846 \pm 0.0002$ |
| GIN | $0.163 \pm 0.004$ | $59.82 \pm 0.33$ | $85.732 \pm 0.023$ | $0.7881 \pm 0.0119$ | $0.2682 \pm 0.0006$ |
| GIN-AK | $0.094 \pm 0.005$ | $67.51 \pm 0.21$ | $86.803 \pm 0.044$ | $0.7829 \pm 0.0121$ | $0.2740 \pm 0.0032$ |
| GIN-AK$^+$ | $\mathbf{0.080 \pm 0.001}$ | $72.19 \pm 0.13$ | $86.850 \pm 0.057$ | $0.7961 \pm 0.0119$ | $\mathbf{0.2930 \pm 0.0044}$ |
| PNA$^*$ | $0.140 \pm 0.006$ | $73.11 \pm 0.11$ | $85.441 \pm 0.009$ | $0.7905 \pm 0.0102$ | $0.2737 \pm 0.0009$ |
| PNA$^*$-AK$^+$ | $0.085 \pm 0.003$ | OOM | OOM | $0.7880 \pm 0.0153$ | $0.2885 \pm 0.0006$ |
| GCN-AK$^+$-S | $0.127 \pm 0.001$ | $71.93 \pm 0.47$ | $86.805 \pm 0.046$ | $0.7825 \pm 0.0098$ | $0.2840 \pm 0.0036$ |
| GIN-AK$^+$-S | $0.083 \pm 0.001$ | $72.39 \pm 0.38$ | $86.811 \pm 0.013$ | $0.7822 \pm 0.0075$ | $0.2916 \pm 0.0029$ |
| PNA$^*$-AK$^+$-S | $0.082 \pm 0.000$ | $\mathbf{74.79 \pm 0.18}$ | $86.676 \pm 0.022$ | $0.7821 \pm 0.0143$ | $0.2880 \pm 0.0012$ |

## 6.3 SCALING UP BY SUBSAMPLING

In some cases, GNN-AK ($^+$)'s overhead leads to OOM, especially for complex models like PNA$^*$ that are resource-demanding. Sampling with SubgraphDrop enables training using practical resources. Notably, GNN-AK$^+$-S models, shown at the end of Table 3, do not compromise and can *even improve* performance as compared to their non-sampled counterpart, in alignment with Dropout's benefits Srivastava et al. (2014). Next we evaluate resource-savings, specifically on ZINC-12K and CIFAR10. Table 4 shows that GIN-AK$^+$-S with varying $R$ provides an effective handle to trade off resources with performance. Importantly, the rate in which performance decays with smaller $R$ is much lower than the rates at which runtime and memory decrease.

Table 4: Resource analysis of SubgraphDrop.

| Dataset | | GIN-AK$^+$-S | | | | | GIN-AK$^+$ | GIN |
|---|---|---|---|---|---|---|---|---|
| | | $R$=1 | $R$=2 | $R$=3 | $R$=4 | $R$=5 | | |
| ZINC-12K | MAE | 0.1216 | 0.0929 | 0.0846 | 0.0852 | 0.0854 | 0.0806 | 0.1630 |
| | Runtime (S/Epoch) | 10.8 | 11.2 | 12.0 | 12.4 | 12.5 | 9.4 | 6.0 |
| | Memory (MB) | 392 | 811 | 1392 | 1722 | 1861 | 1911 | 124 |
| CIFAR10 | ACC | 71.68 | 72.07 | 72.39 | 72.20 | 72.32 | 72.19 | 59.82 |
| | Runtime (S/Epoch) | 80.7 | 89.1 | 100.5 | 110.9 | 119.7 | 241.1 | 55.0 |
| | Memory (MB) | 2576 | 4578 | 6359 | 8716 | 10805 | 30296 | 801 |

## 6.4 ABLATION STUDY

We present ablation results on various structural components of GNN-AK($^+$) in Appendix A.11. Table 6 shows the performance of GIN-AK$^+$ for varying size egonets with $k$. Table 7 illustrates the added benefit of various encodings and D2C feature. Table 8 extensively studies the effect of context encoding and D2C in GNN-AK$^+$, as they are not explained by Subgraph-1-WL$^*$. Table 9 studies the effect of base model's depth (or expressiveness) for GNN-AK$^+$ with and without D2C.

## 7 CONCLUSION

Our work introduces a new, general-purpose framework called GNN-As-Kernel (GNN-AK) to uplift the expressiveness of any GNN, with the key idea of employing a base GNN as a kernel on induced subgraphs of the input graph, generalizing from the star-pattern aggregation of classical MPNNs. Our approach provides an expressiveness and performance boost, while retaining practical scalability of MPNNs—a highly sought-after middle ground between the two regimes of scalable yet less-expressive MPNNs and high-expressive yet practically infeasible and poorly-generalizing $k$-order designs. We theoretically studied the expressiveness of GNN-AK, provided a concrete design and the more powerful GNN-AK$^+$, introducing SubgraphDrop for shrinking runtime and memory footprint. Extensive experiments on both simulated and real-world benchmark datasets empirically justified that GNN-AK($^+$) (i) uplifts base GNN expressiveness for multiple base GNN choices (e.g. over 1&2-WL for MPNNs, and over 3-WL for PPGN), (ii) which translates to performance gains with SOTA results on graph-level benchmarks, (iii) while retaining scalability to practical graphs.

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

# A APPENDIX

## A.1 ADDITIONAL RELATED WORK

**Improving Expressiveness of GNNs:** Several works other than those mentioned in Sec.1 tackle expressive GNNs. Murphy et al. (2019) achieve universality by summing permutation-sensitive functions across a combinatorial number of permutations, limiting feasibility. Dasoulas et al. (2020) adds node indicators to make them distinguishable, but at the cost of an invariant model, while Vignac et al. (2020) further addresses the invariance problem, but at the cost of quadratic time complexity. Corso et al. (2020) generalizes MPNN's default sum aggregator, but is still limited by 1-WL. Beani et al. (2021) generalizes spatial and spectral aggregation with $>$1-WL expressiveness, but using expensive eigendecomposition. Recently, Bodnar et al. (2021b) introduce MPNNs over simplicial complexes that shares similar expressiveness as GNN-AK. Ying et al. (2021) studies transformer with above 1-WL expressiveness. Azizian & Lelarge (2021) surveys GNN expressiveness work.

**Connections to CNN and $k$-WL:** GNN-AK has a similar convolutional structure as CNN, and in fact historically many spatial GNNs are inspired by CNN; see Wu et al. (2020) for a detailed survey. The non-Euclidean nature of graphs makes such generalizations non-trivial. MoNet (Monti et al., 2017) introduces pseudo-coordinates for nodes, while PatchySAN (Niepert et al., 2016) learns to order and truncate neighboring nodes for convolution purposes. However, both methods aim to mimic the formulation of the CNN without admitting the inherent difference between graphs and images. In contrast, GNN-AK, generalizes CNN to graphs with a base GNN kernel, similar to how a CNN kernel encodes image patches. GNN-AK also shares connections with two variants of $k$-WL test algorithms: depth-$k$ 1-dim WL (Cai et al., 1992; Weisfeiler, 1976) and deep WL (Arvind et al., 2020). The former recursively applies 1-WL to all size-$k$ subgraphs, with slightly weaker expressiveness than $k$-WL, and the latter reduces the number of such subgraphs for the $k$-WL test. Instead of working on all $O(n^k)$ size-$k$ subgraphs, we keep linear scalability by only applying 1-WL-equivalent MPNNs to $O(n)$ rooted subgraphs.

## A.2 SAMPLING BY SUBGRAPHDROP

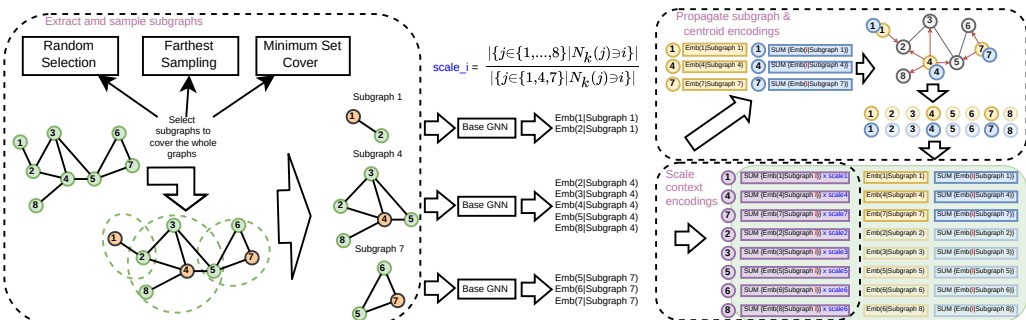

Figure 2: GNN-AK-S with SubgraphDrop used in training. GNN-AK-S first extracts subgraphs and subsamples $m \ll |\mathcal{V}|$ subgraphs to cover each node at least $R$ times with multiple strategies. The base GNN is applied to compute all intermediate node embeddings in selected subgraphs. Context encodings are scaled to match evaluation. Subgraph and centroid encodings initially only exist for root nodes of selected subgraphs, and are propagated to estimate those of other nodes.

The descriptions of subgraph sampling strategies are as follows. Fig. 2 shows an overview of SubgraphDrop.

- **Random sampling** selects subgraphs randomly until every node is covered $\geq R$ times.
- **Farthest sampling** selects subgraphs iteratively, starting at a random one and greedily selecting each subsequent one whose root node is farthest w.r.t. shortest path distance from those of already selected subgraphs, until every node is covered $\geq R$ times.
- **Min-set-cover sampling** initially selects a subgraph randomly, and follows the greedy minimum set cover algorithm to iteratively select the subgraph containing the maximum number of uncovered nodes, until every node is covered $\geq R$ times.

## A.3 PROOF OF THEOREM 1

*Proof.* (Xu et al., 2018) proved that with sufficient number of layers and all injective functions, MPNN is as powerful as 1-WL. Then Eq.2 outputs different vectors for two graphs iff Subgraph-1-WL* encodes different labels with $c_v^{(t+1)} = $ 1-WL$(G^{(t)}[\mathcal{N}_k(v)])$. With POOL in Eq.2 also to be injective, MPNN-AK outputs different vectors iff Subgraph-1-WL* outputs different fingerprints for two graphs. Then MPNN-AK is as powerful as Subgraph-1-WL*. ☐

## A.4 PROOF OF THEOREM 2

*Proof.* We first prove that if two graphs are identified as isomorphic by Subgraph-1-WL*, they are also determined as isomorphic by 1-WL. Then we present a pair of non-isomorphic graphs that can be distinguished by Subgraph-1-WL* but not by 1-WL. Together these two imply that Subgraph-1-WL* is strictly more powerful than 1-WL. Comparing with 2-WL can be concluded from the fact that 1-WL and 2-WL are equivalent in expressiveness (Maron et al., 2019a). In the proof we use 1-hop egonet subgraphs for Subgraph-1-WL*.

Assume graphs $G$ and $H$ have the same number of nodes (otherwise easily determined as non-isomorphic) and are two non-isomorphic graphs but Subgraph-1-WL* determines them as isomorphic. Then for any iteration $t$, set $\left\{ \text{1-WL}(G^{(t)}[\mathcal{N}_1(v)])|v \in \mathcal{V}_G \right\}$ is the same as set $\left\{ \text{1-WL}(H^{(t)}[\mathcal{N}_1(v)])|v \in \mathcal{V}_H \right\}$. Then there existing an ordering of nodes $v_1^G, ..., v_n^G$ and $v_1^H, ..., v_N^H$ with $N = |\mathcal{V}_G| = |\mathcal{V}_H|$, such that for any node order $i = 1, ..., N$, 1-WL$(G^{(t)}[\mathcal{N}_1(v_i^G)]) = $ 1-WL$(H^{(t)}[\mathcal{N}_1(v_i^H)])$. This implies that structure $G[\mathcal{N}_1(v_i^G)]$ and $H[\mathcal{N}_1(v_i^H)]$ are not distinguishable by 1-WL. Hence $Star(v_i^G)$ and $Star(v_i^H)$ are hashed to the same label otherwise the 1-WL that includes the hashed result of $Star(v_i^G)$ and $Star(v_i^H)$ can also distinguish $G[\mathcal{N}_1(v_i^G)]$ and $H[\mathcal{N}_1(v_i^H)]$. Then for any iteration $t$ and any node with order $i$, HASH$\left( Star(v_i^{G^{(t)}}) \right) = $ HASH$\left( Star(v_i^{H^{(t)}}) \right)$ implies that 1-WL fails in distinguishing $G$ and $H$. In fact if we replace the 1-WL hashing function in Subgraph-1-WL* to a stronger version HASH$\left( \left\{ \text{HASH}(Star(v^{G^{(t)}})), \text{1-WL}(G^{(t)}[\mathcal{N}_1(v)]) \right\} \right)$, this directly implies the above statement.

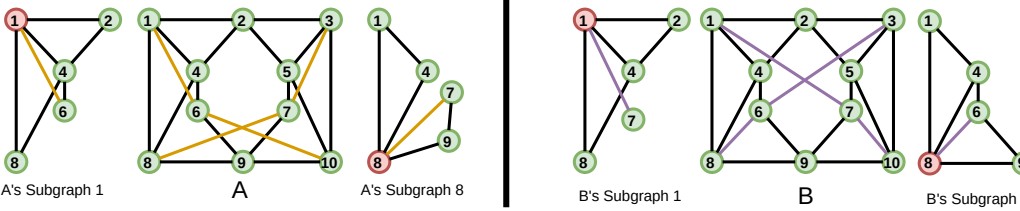

Figure 3: Two 4-regular graphs that cannot be distinguished by 1-WL. Colored edges are the difference between two graphs. Two 1-hop egonets are visualized while all other rooted egonets are ignored as they are same across graph $A$ and graph $B$.

In Figure 3, two 4-regular graphs are presented that cannot be distinguished by 1-WL but can be distinguished by Subgraph-1-WL*. We visualize the 1-hop egonets that are structurally different among graphs $A$ and $B$. It's easy to see that $A$'s egonet $A[\mathcal{N}_1(1)]$ and $B$'s egonet $B[\mathcal{N}_1(1)]$ can be distinguished by 1-WL, as degree distribution is not the same. Hence, $A$ and $B$ can be distinguished by Subgraph-1-WL*. ☐

## A.5 PROOF OF THEOREM 3

*Proof.* We prove by showing a pair of 3-WL failed non-isomorphic graphs can be distinguished by Subgraph-1-WL (see definition of "no less powerful" in Zhengdao et al. (2020): we call A is no/not less powerful than B if there exists a pair of non-isomorphic graphs that cannot be distinguished by B but can be distinguished by A.), assuming HASH(·) is 3-WL discriminative. Figure 4 shows two strongly regular graphs that can not be distinguished by 3-WL (any strongly regular graphs are not distinguishable by 3-WL (Arvind et al., 2020)), along with their 1-hop egonet rooted subgraphs.

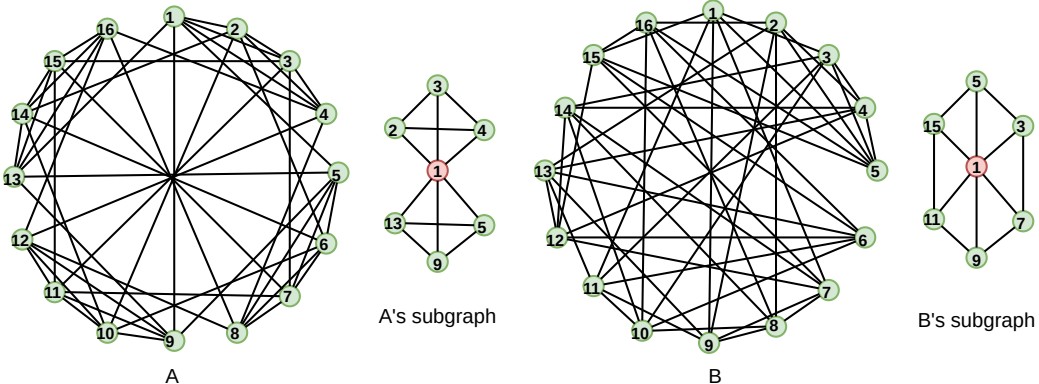

Figure 4: Two non-isomorphic strongly regular graphs that cannot be distinguished by 3-WL.

Notice that all 1-hop egonet rooted subgraphs in $A$ (also in $B$) are the same, resulting that Subgraph-1-WL can successfully distinguish $A$ and $B$ if HASH can distinguish the showed two 1-hop egonets. Now we prove that 3-WL can distinguish these two *subgraphs*. 3-WL constructs a coloring of 3-tuples of all vertices in a graph, and uses the histogram of colors of all $k$-tuples as fingerprint of the graph. Then, different 3-tuples correspond to different colors or bins in the histogram. As a triangle is a unique type of 3-tuple, at iteration 0 of 3-WL, the histogram of all 3-tuples counts the number of triangles in the graph. Notice that $A$'s subgraph has $2 \times \binom{4}{3} = 8$ triangles, and $B$'s subgraph contains 6 triangles. This implies that even at iteration 0, 3-WL can distinguish these two subgraphs. Hence, when HASH$(\cdot)$ is 3-WL expressive, Subgraph-1-WL can distinguish $A$ and $B$. Therefore there exists a pair of 3-WL failed non-isomorphic graphs that can be distinguished by Subgraph-1-WL. □

### A.6 Proof of Theorem 4

*Proof.* For any $k \geq 3$, Cai et al. (1992) provided a scheme to construct a pair of $k$-WL-failed non-isomorphic graphs based on a base graph $G$ with separator size $k + 1$ (more specifically, $G$ can be chosen as a degree-3 regular graph with separator size $k + 1$, see Corollary 6.5 in Cai et al. (1992)).

We describe the proposed scheme of constructing the two graphs $A$ and $B$ briefly, for details see Section 6 in Cai et al. (1992). At high level, this scheme first constructs $A$ by replacing every node (with degree $d$) in $G$ by a carefully designed graph $X_d$, and then rewires two edges in $A$ to its non-isomorphic counter graph $B$. Specifically, the small graph $X_d = (\mathcal{V}_d, \mathcal{E}_d)$ is defined as follows.

$$\mathcal{V}_d = A_d \cup B_d \cup M_d, \text{ where } A_d = \{a_i | 1 \leq i \leq d\}, B_d = \{b_i | 1 \leq i \leq d\},$$
$$\text{and } M_d = \{m_S | S \subseteq \{1, ..., d\}, |S| \text{ is even}\} \tag{11}$$
$$\mathcal{E}_d = \{(m_S, a_i) | i \in S\} \cup \{(m_S, b_i) | i \notin S\}$$

Hence each vertex $v$ with degree $d$ in $G$ is replaced by a graph $X(v) = X_d$ of size $2^{d-1} + 2d$, including a "middle" section $M_d$ of size $2^{d-1}$ and $d$ pairs of vertices $(a_i, b_i)$ representing "endpoints" of each edge incident with $v$ in the original graph $G$. Let $(u, v) \in \mathcal{E}(G)$, then two small graphs $X(v)$ and $X(u)$ in $A$ are connected by the following. Let $(a_{u,v}, b_{u,v}), (a_{v,u}, b_{v,u})$ be the pair of vertices associated with $(u, v)$ in $X(u)$ and $X(v)$ respectively, then $a_{u,v}$ is connected to $a_{v,u}$, and $b_{u,v}$ is connected to $b_{v,u}$. After constructing $A$, $B$ is modified from $A$ by rewiring a single edge $(u, v)$ in the original graph $G$. Specifically, now in $B$, $a_{u,v}$ is connected to $b_{v,u}$ and $b_{u,v}$ is connected to $a_{v,u}$. Cai et al. (1992) proved that $A$ and $B$ are non-isomorphic, and when base graph $G$ is a degree-3 regular graph with separator size $k + 1$, $A$ and $B$ are not distinguishable by $k$-WL. See a visualization in Figure 5.

We provide several useful lemmas related to graph $X_d$ in the following, which are then used in proving Theorem 4.

**Lemma 5** (Cai et al. (1992)). *$X_d$ has exactly $2^{d-1}$ automorphisms, and each automorphism is determined by interchanging $a_i$ and $b_i$ for each $i$ in some subset $S \subseteq \{1, ..., d\}$ of even cardinality.*

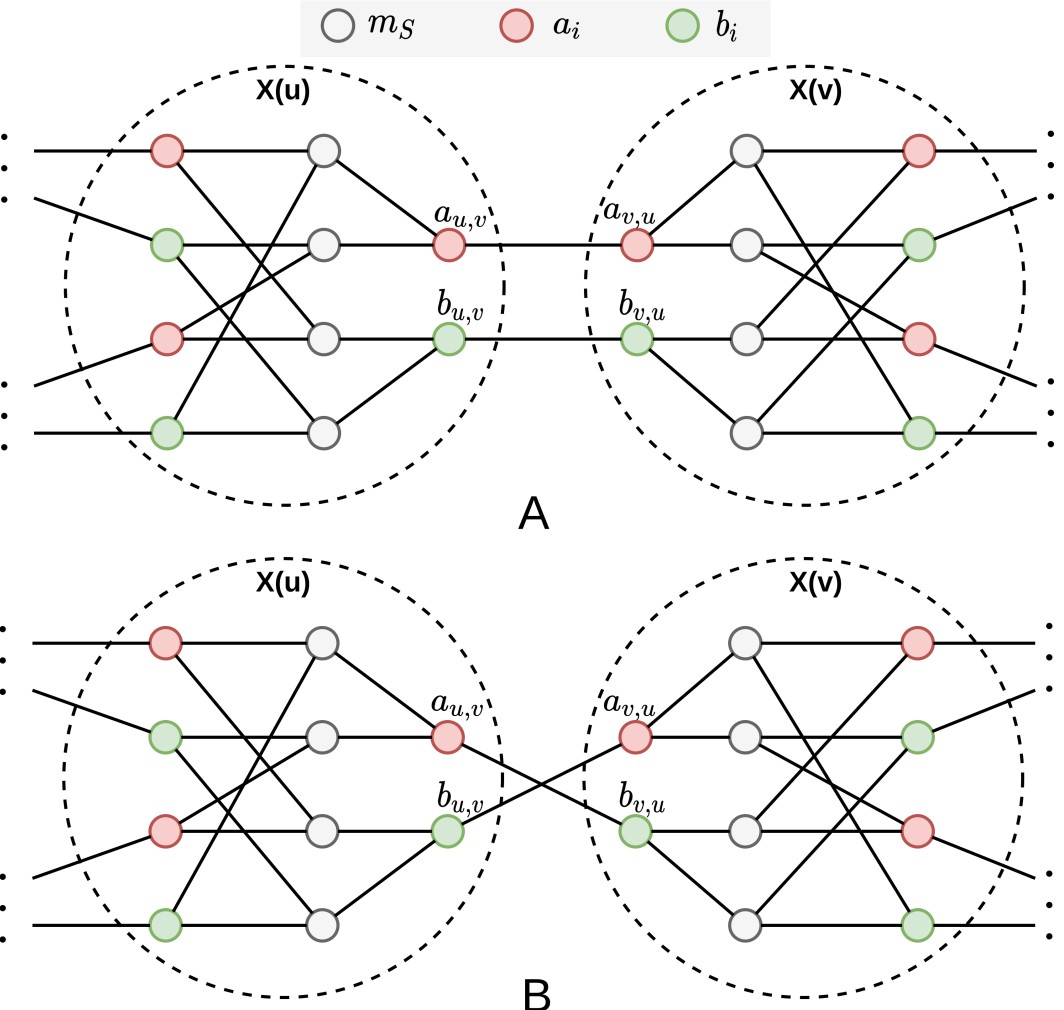

Figure 5: A pair of CFI graphs ($A$ and $B$) zoomed at the (rewired) edge $(u, v)$ of a base graph. The base graph is a degree-3 regular graph with separator size $k + 1$ for $k$-WL-failed case.

Lemma 5 is directly taken from Lemma 6.1 in Cai et al. (1992).

**Lemma 6.** *The shortest path distance between $a_i$ and $b_i$ in $X_d$ for any $i$ is exactly 4, and diameter of $X_d$ is 4.*

*Proof.* Based on the construction shown in Eq.11, for any $i$, $a_i$ and $b_i$ are not directly connected, and any middle nodes share no connection. This implies the shortest path between $a_i$ and $b_i$ is larger than 3. Let $m_S$ be one middle node connected with $a_i$. We construct a new set $S' = \{p, q\}$, with $p \neq i, p \in S$ and $q \notin S$, then $m_{S'}$ connects with $b_i$ based on Eq.11. Additionally, $a_p$ connects to both $m_S$ and $m_{S'}$, which implies the shortest path distance between $a_i$ and $b_i \leq 4$. Hence, the shortest path distance between $a_i$ and $b_i$ is exactly 4. To show the diameter of $X_d$ is 4, it's easy to observe that distance($a_i, b_i$) only decreases when replacing $a_i$ with other nodes in $X_d$ (or reverse, replacing $b_i$ with other nodes in $X_d$). $\square$

**Lemma 7.** *For any $k \geq 1$, $k$-egonets of $a_i$ and $b_i$ in $X_d$ are isomorphic.*

*Proof.* When $k \geq 4$, $X_d[\mathcal{N}_k(a_i)] = X_d[\mathcal{N}_k(b_i)] = X_d$ based on Lemma 6. Now for $1 \leq k \leq 3$, we analyze the $k$-hop egonets of $a_i$ and $b_i$ sequentially. First, $X_d[\mathcal{N}_1(a_i)]$ and $X_d[\mathcal{N}_1(b_i)]$ are two depth-1 trees with equal number of leaves (middle nodes in $X_d$), and of course are isomorphic.

Similarly, $X_d[\mathcal{N}_2(a_i)]$ and $X_d[\mathcal{N}_2(b_i)]$ are two depth-2 trees with equal number of depth-1 nodes and equal number of leaves, which are also isomorphic. Last, $X_d[\mathcal{N}_3(a_i)]$ and $X_d[\mathcal{N}_3(b_i)]$ are the graphs by removing $b_i$ from $X_d$ and $a_i$ from $X_d$, respectively. It's trivial to observe that they are also isomorphic. Combining all cases proves the Lemma. □

To prove the main theorem, we visualize the two graphs A and B zoomed at edge $(u, v)$ of base graph in Figure 5 to help understanding. To prove that Subgraph-1-WL with $t \leq 4$-egonet (we use $t$ to prevent notation conflict) cannot distinguish $A$ and $B$, we mainly analyze the subgraph around $a_{u,v}$ for both graph $A$ and $B$. For other subgraphs with different root nodes within $t$ shortest-path-distance around nodes $a_{u,v}, a_{v,u}, b_{u,v}, b_{v,u}$, they follow exactly the same argument. For all subgraphs with other root nodes, it's trivial to observe that there is no difference among two subgraphs in $A$ and $B$.

We introduce some notation first. Let $A[\mathcal{N}_t(a_{u,v})]$ and $B[\mathcal{N}_t(a_{u,v})]$ be the two $t$-egonet subgraphs around $a_{u,v}$ in $A$ and $B$ respectively. Let $A^{left}[\mathcal{N}_t(a_{u,v})]$ be the "left" part graph (visualized in Figure 5, the left side of $a_{u,v}$, include $a_{u,v}$) of $A[\mathcal{N}_t(a_{u,v})]$, and $A^{right}[\mathcal{N}_t(a_{u,v})]$ be the "right" part graph (visualized in Figure 5, the right side of $a_{u,v}$, excluding $a_{u,v}$) of $A[\mathcal{N}_t(a_{u,v})]$. Then $A[\mathcal{N}_t(a_{u,v})]$ is reconstructed by connecting $A^{left}[\mathcal{N}_t(a_{u,v})]$ and $A^{right}[\mathcal{N}_t(a_{u,v})]$ with a single edge $(a_{u,v}, a_{v,u})$. $B^{left}[\mathcal{N}_t(a_{u,v})]$ and $B^{right}[\mathcal{N}_t(a_{u,v})]$ are defined in the same way, such that $B[\mathcal{N}_t(a_{u,v})]$ is reconstructed by connecting $(a_{u,v}, b_{v,u})$ in $B$.

When $t \leq 4$, we show that there exists an isomorphism that maps $A[\mathcal{N}_t(a_{u,v})]$ to $B[\mathcal{N}_t(a_{u,v})]$. We construct this isomorphism by constructing an isomorphism between $A^{left}[\mathcal{N}_t(a_{u,v})]$ and $B^{left}[\mathcal{N}_t(a_{u,v})]$ and an isomorphism between $A^{right}[\mathcal{N}_t(a_{u,v})]$ and $B^{right}[\mathcal{N}_t(a_{u,v})]$ first. Trivially, the current node ordering in $A^{left}[\mathcal{N}_t(a_{u,v})]$ and $B^{left}[\mathcal{N}_t(a_{u,v})]$ already defines an isomorphism among them, that is, mapping any $a_i$ ($b_i, m_S$) in $A^{left}[\mathcal{N}_t(a_{u,v})]$ to $a_i$ ($b_i, m_S$) in $B^{left}[\mathcal{N}_t(a_{u,v})]$. To find an isomorphism between $A^{right}[\mathcal{N}_t(a_{u,v})]$ and $B^{right}[\mathcal{N}_t(a_{u,v})]$, one can easily observe that for $t \leq 3$, $A^{right}[\mathcal{N}_t(a_{u,v})]$ can be 1-to-1 mapped (with $a_{v,u}$ in $A$ being mapped to $a_i$) to $X_3[\mathcal{N}_{t-1}(a_i)]$ and $B^{right}[\mathcal{N}_t(a_{u,v})]$ can be 1-to-1 mapped (with $b_{v,u}$ in $A$ being mapped to $b_i$) to $X_3[\mathcal{N}_{t-1}(b_i)]$ for some $i$. When $t = 4$, $A^{right}[\mathcal{N}_t(a_{u,v})]$ is mapped to $X_3[\mathcal{N}_3(a_i)]$ with additional 2 nodes and $B^{right}[\mathcal{N}_t(a_{u,v})]$ is mapped to $X_3[\mathcal{N}_3(b_i)]$ with additional 2 nodes. Then applying Lemma 7, when $t \leq 4$ we can construct an isomorphism between $A^{right}[\mathcal{N}_t(a_{u,v})]$ and $N^{right}[\mathcal{N}_t(a_{u,v})]$ with $a_{v,u}$ in $A$ being mapped to $b_{v,u}$ in $B$. We remark that when $t = 4$, the two additional nodes outside $X_3[\mathcal{N}_3(a_i)]$ and $X_3[\mathcal{N}_3(b_i)]$ does not affecting applying Lemma 7. Last, the combination of the isomorphism between $A^{left}[\mathcal{N}_t(a_{u,v})]$ and $B^{left}[\mathcal{N}_t(a_{u,v})]$ and the isomorphism between $A^{right}[\mathcal{N}_t(a_{u,v})]$ and $B^{right}[\mathcal{N}_t(a_{u,v})]$ is actually a new isomorphism between $A[\mathcal{N}_t(a_{u,v})]$ and $B[\mathcal{N}_t(a_{u,v})]$, as the edge $(a_{u,v}, a_{v,u})$ is mapped to edge $(a_{u,v}, b_{v,u})$. Thus, when $t \leq 4$, $A[\mathcal{N}_t(a_{u,v})]$ and $B[\mathcal{N}_t(a_{u,v})]$ are isomorphic and $\text{HASH}(A[\mathcal{N}_t(a_{u,v})]) = \text{HASH}(B[\mathcal{N}_t(a_{u,v})])$ no matter what $\text{HASH}(\cdot)$ function is used. Applying the same argument to all other pairs of subgraphs in $A$ and $B$ implies that the histogram encodings of $A$ and $B$ are the same. Hence Subgraph-1-WL with $t \leq 4$ cannot distinguish the two graphs $A$ and $B$ that are $k$-WL-failed for any $k \geq 3$. □

We remark that the theorem doesn't imply that Subgraph-1-WL with $t \geq 5$-egonet can distinguish $A$ and $B$. This should depend on the base graph. We give an informal sketch. Suppose that for $t \geq 5$, the left part $A^{left}[\mathcal{N}_t(a_{u,v})]$ and right part $A^{right}[\mathcal{N}_t(a_{u,v})]$ are only connected through $(a_{u,v}, a_{v,u})$, that is, removing $(a_{u,v}, a_{v,u})$ from $A[\mathcal{N}_t(a_{u,v})]$ resulting in $A^{left}[\mathcal{N}_t(a_{u,v})]$ and $A^{right}[\mathcal{N}_t(a_{u,v})]$ being disconnected. Then we can apply Lemma 5 multiple times with mapping some $a_{v,u}$ in A to $b_{v,u}$ in B and $b_{v,u}$ in A to $a_{v,u}$ in B, until this kind of switches reach the boundary of $A[\mathcal{N}_t(a_{u,v})]$, resulting in an isomorphism between $A[\mathcal{N}_t(a_{u,v})]$ and $B[\mathcal{N}_t(a_{u,v})]$. We refer the reader to Lemma 6.2 in Cai et al. (1992), which also uses this line of argument.

### A.7    PROOF OF PROPOSITION 1

*Proof.* The proof is by contradiction. Let $G$ and $H$ be two non-isomorphic graphs and $|V_G| = |V_H|$ (if graph sizes are different then Subgraph-1-WL can trivially distinguish them). Now suppose that Prop. 1 is incorrect, i.e. Subgraph-1-WL cannot distinguish $G$ and $H$ even provided that the two conditions (1) and (2) are satisfied. Then, for any iteration of Subgraph-1-WL, $G$ and $H$ would have the *same* histogram of subgraph colors. Now we focus on iteration 0. Formally, let color

$c_{v,i}^G = \text{HASH}(G[\mathcal{N}_k(v_i^G)])$ and $c_{v,i}^H = \text{HASH}(H[\mathcal{N}_k(v_i^H)])$ for a node order $v$ ($v_i^G$ maps index $i$ to a node in $G$). According to Condition (1): for any node reordering $v_1^G, ..., v_{N_G}^G$ and $v_1^H, ..., v_{N_H}^H$, $\exists i \in [1, \max(N_G, N_H)]$ that $G[\mathcal{N}_k(v_i^G)]$ and $H[\mathcal{N}_k(v_i^G)]$ are non-isomorphic, then we know that $\exists i$ that $G[\mathcal{N}_k(v_i^G)]$ and $H[\mathcal{N}_k(v_i^H)]$ are non-isomorphic, hence $c_{v,i}^G \neq c_{v,i}^H$ since by Condition (2) HASH can distinguish these two subgraphs. However as two graphs have the same histogram of subgraph colors, there must be a $j \neq i$ such that $c_{v,i}^G = c_{v,j}^H$ and $c_{v,j}^G = c_{v,i}^H$. Then we can create a new node order $m$ by swapping $v_i^G$ and $v_j^G$, resulting $c_{m,i}^G = c_{m,i}^H$ and $c_{m,j}^G = c_{m,j}^H$. This process can be repeated until having a new node order $w$ such that $\forall i \in 1, ..., |V_G|, c_{w,i}^G = c_{w,i}^H$. As HASH is discriminative enough according to Condition (2), this implies $\forall i \in 1, ..., |V_G|, G[\mathcal{N}_k(w_i^G)]$ and $H[\mathcal{N}_k(w_i^H)]$ are isomorphic, which contradicts with Condition (1). Thus, Prop. 1 must be true. $\square$

## A.8 Proof of Proposition 2

*Proof.* When a pair of non-isomorphic graphs $G$ and $H$ cannot be distinguished by GNN-AK$^+$, for any layer $l$, the histogram of $h_v^{(l))}$ in $G$ and $H$ in Eq. 7 should be the same. Which implies that for any layer $l$, the histogram of $h_v^{(l))}$ in $G$ and $H$ in Eq. 4 should be the same. Then GNN-AK cannot distinguish $G$ and $H$. So for any pair of non-isomorphic graphs, GNN-AK cannot distinguish them if GNN-AK$^+$ cannot distinguish them. Thus GNN-AK$^+$ is more powerful than GNN-AK. $\square$

## A.9 Dataset Description & Statistics

Table 5: Dataset statistics.

| Dataset | Task Semantic | # Cls./Tasks | # Graphs | Ave. # Nodes | Ave. # Edges |
|---|---|---|---|---|---|
| EXP | Distinguish 1-WL failed graphs | 2 | 1200 | 44.4 | 110.2 |
| SR25 | Distinguish 3-WL failed graphs | 15 | 15 | 25 | 300 |
| CountingSub. | Regress num. of substructures | 4 | 1500 / 1000 / 2500 | 18.8 | 62.6 |
| GraphProp. | Regress global graph properties | 3 | 5120 / 640 / 1280 | 19.5 | 101.1 |
| ZINC-12K | Regress molecular property | 1 | 10000 / 1000 / 1000 | 23.1 | 49.8 |
| CIFAR10 | 10-class classification | 10 | 45000 / 5000 / 10000 | 117.6 | 1129.8 |
| PATTERN | Recognize certain subgraphs | 2 | 10000 / 2000 / 2000 | 118.9 | 6079.8 |
| MolHIV | 1-way binary classification | 1 | 32901 / 4113 / 4113 | 25.5 | 54.1 |
| MolPCBA | 128-way binary classification | 128 | 350343 / 43793 / 43793 | 25.6 | 55.4 |
| MUTAG | Recognize mutagenic compounds | 2 | 188 | 17.93 | 19.79 |
| PTC-MR | Classify chemical compounds | 2 | 344 | 14.29 | 14.69 |
| PROTEINS | Classify Enzyme & Non-enzyme | 2 | 1113 | 39.06 | 72.82 |
| NCI1 | Classify molecular | 2 | 4110 | 29.87 | 32.30 |
| IMDB-B | Classify movie | 2 | 1000 | 19.77 | 96.53 |
| RDT-B | Classify reddit thread | 2 | 2000 | 429.63 | 497.75 |

## A.10 Experimental Setup Details

To reduce the search space, we search hyperparameters in a two-phase approach: First, we search common ones (hidden size from [64, 128], number of layers $L$ from [2,4,5,6], (sub)graph pooling from [SUM, MEAN] for each dataset using GIN based on validation performance, and fix it for any other GNN and GNN-AK($^+$). While hyperparameters may not be optimal for other GNN models, the evaluation is fair as there is no benefit for GNN-AK($^+$). Next, we search GNN-AK($^+$) exclusive ones (encoding types) over validation set using GIN-AK($^+$) and keep them fixed for other GNN-AK($^+$). We use a 1-layer base model for GNN-AK($^+$), with exceptions that we tune number of layers of base model (while keeping total number of layers fixed) for GNN-AK in simulation datasets (presented in Table 1). We use 3-hop egonets for GNN-AK($^+$), with exceptions that CIFAR10 uses 2-hop egonet due to memory constraint; PATTERN and RDT-B use random walk based subgraph with walk length=10 and repeat times=5 as their graphs are dense. For GNN-AK($^+$)-S, $R = 3$ is set as default. We use farthest sampling for molecular datasets ZINC, MolHIV, and MolPCBA; to speed up further, random sampling is used for CIFAR10 whose graphs are $k$-NN graphs; min-set-cover sampling is used for PATTERN to adapt random walk based subgraph. We use Batch Normalization and ReLU activation in all models. For optimization we use Adam with learning rate 0.001 and

weight decay 0. All experiments are repeated 3 times to calculate mean and standard derivation. All experiments are conducted on RTX-A6000 GPUs.

### A.11 Ablation Study

We present ablation results on various structural components of GNN-AK. Table 6 shows the performance of GIN-AK for varying size egonets with $k$. We find that larger subgraphs tend to yield improvement, although runtime-performance trade-off may vary by dataset. Notably, simply 1-egonets are enough for CIFAR10 to uplift performance of the base GIN considerably.

Table 6: Effect of various $k$-egonet size.

| $k$ of GIN-AK$^+$ | ZINC-12K | CIFAR10 |
|---|---|---|
| GIN | $0.163 \pm 0.004$ | $59.82 \pm 0.33$ |
| $k = 1$ | $0.147 \pm 0.006$ | $71.37 \pm 0.28$ |
| $k = 2$ | $0.120 \pm 0.005$ | $72.19 \pm 0.13$ |
| $k = 3$ | $0.080 \pm 0.001$ | OOM |

Table 7: Effect of various encodings

| Ablation of GIN-AK$^+$ | ZINC-12K | CIFAR10 |
|---|---|---|
| Full | $0.080 \pm 0.001$ | $72.19 \pm 0.13$ |
| w/o Subgraph encoding | $0.086 \pm 0.001$ | $67.76 \pm 0.29$ |
| w/o Centroid encoding | $0.084 \pm 0.003$ | $72.20 \pm 0.96$ |
| w/o Context encoding | $0.088 \pm 0.003$ | $69.25 \pm 0.30$ |
| w/o Distance-to-Centroid | $0.085 \pm 0.001$ | $71.91 \pm 0.22$ |

Next Table 7 illustrates the added benefit of various node encodings. Compared to the full design, eliminating any of the subgraph, centroid, or context encodings (Eq.s (3)–(5)) yields notably inferior results. Encoding without distance awareness is also subpar. These justify the design choices in our framework and verify the practical benefits of our design.

As GNN-AK$^+$ is not directly explained by Subgraph-1-WL$^*$ (though D2C is supported by Subgraph-1-WL, by strengthening HASH), we conduct additional ablation studies over GNN-AK$^+$ with different combinations of removing context encoding and D2C, shown in Table 8. Note that all experiments in Table 8 are using a 1-layer base GIN model. We summarize the observations as follows.

- For real-world datasets (ZINC-12K, CIFAR10), We observe that the largest performance improvement over base model comes from wrapping base GNN with Eq.2 (the performance of GIN-AK), while adding context encoding and D2C monotonically improves performance of GNN-AK$^+$.
- For substructure counting and graph property regression tasks, we observe D2C significantly increases the performance of GIN-AK$^+$ w/o Ctx (supported by the theory of Subgraph-1-WL where the HASH is required to be injective). Specifically, the cost-free D2C feature enhances the expressiveness of the base 1-layer GIN model (similar to the benefit of distance encoding shown in Li et al. (2020)), resulting in a more expressive GNN-AK$^+$, which lies between Subgraph-1-WL and Subgraph-1-WL$^*$. We leave the exact theoretical analysis of D2C's expressiveness benefit in future work. Notice that GIN-AK improves the performance over the base model but not in a large margin, we next show this is due to the insufficiency of the number of layers of base GIN.

Table 8: Study GNN-AK without context encoding (Ctx) and without distance-to-centroid (D2C). Base model is **1-layer** GIN for all methods.

| Method | ZINC-12K (MAE) | CIFAR10 (ACC) | EXP (ACC) | SR25 (ACC) | Counting Substructures (MAE) | | | | Graph Properties ($\log_{10}$(MAE)) | | |
|---|---|---|---|---|---|---|---|---|---|---|---|
| | | | | | Triangle | Tailed Tri. | Star | 4-Cycle | IsConnected | Diameter | Radius |
| GIN | 0.163 | 59.82 | 50% | 6.67% | 0.3569 | 0.2373 | 0.0224 | 0.2185 | -1.9239 | -3.3079 | -4.7584 |
| GIN-AK | 0.094 | 67.51 | 100% | 6.67% | 0.2311 | 0.1805 | 0.0207 | 0.1911 | -1.9574 | -3.6925 | -5.0574 |
| GIN-AK$^+$ w/o Ctx | 0.088 | 69.25 | 100% | 6.67% | 0.0130 | 0.0108 | 0.0177 | 0.0131 | -2.7083 | -3.9257 | -5.2784 |
| GIN-AK$^+$ w/o D2C | 0.085 | 71.91 | 100% | 6.67% | 0.1746 | 0.1449 | 0.0193 | 0.1467 | -2.0521 | -3.6980 | -5.0984 |
| GIN-AK$^+$ | 0.080 | 72.19 | 100% | 6.67% | 0.0123 | 0.0112 | 0.0150 | 0.0126 | -2.7513 | -3.9687 | -5.1846 |

According to Prop. 1, the base model must be discriminative enough such that $\text{HASH}(G[\mathcal{N}_k(v_i^G)]) \neq \text{HASH}(H[\mathcal{N}_k(v_i^H)])$, for Subgraph-1-WL with $k$-egonet to enjoy expressiveness benefits. In addition to using D2C to increase the expressiveness of base model, another way is to practically increase the number of layers of the base model (akin to increasing the number of iterations of 1-WL, as in the definition of Subgraph-1-WL$^*$). We study the effect of base model's number of layers in GNN-AK$^+$, with and without D2C in Table 9.

- Firstly, GNN-AK$^+$ with D2C is insensitive to the depth of base model, with 1-layer base model being enough to achieve great performance in counting substructures and the best performance

Table 9: Study the effect of base model's number of layers while keeping total number of layers in GNN-AK fixed. Different effect is observed for GNN-AK and GNN-AK without D2C.

| Method | GIN-AK's #Layers | Base GIN's #Layers | Counting Substructures (MAE) | | | | Graph Properties ($\log_{10}$(MAE)) | | |
|---|---|---|---|---|---|---|---|---|---|
| | | | Triangle | Tailed Tri. | Star | 4-Cycle | IsConnected | Diameter | Radius |
| GIN | 0 | 6 | 0.3569 | 0.2373 | 0.0224 | 0.2185 | -1.9239 | -3.3079 | -4.7584 |
| GIN-AK | 6 | 1 | 0.2311 | 0.1805 | 0.0207 | 0.1911 | -1.9574 | -3.6925 | -5.0574 |
| | 3 | 2 | 0.1556 | 0.1275 | 0.0172 | 0.1419 | -1.9134 | -3.7573 | -5.0100 |
| | 2 | 3 | 0.1064 | 0.0819 | 0.0168 | 0.1071 | -1.9259 | -3.7243 | -4.9257 |
| | 1 | 6 | 0.0934 | 0.0751 | 0.0216 | 0.0726 | -1.9916 | -3.6555 | -4.9249 |
| GIN-AK$^+$ w/o D2C | 6 | 1 | 0.1746 | 0.1449 | 0.0193 | 0.1467 | -2.0521 | -3.6980 | -5.0984 |
| | 3 | 2 | 0.1244 | 0.1052 | 0.0219 | 0.1121 | -2.1538 | -3.7305 | -4.9250 |
| | 2 | 3 | 0.1021 | 0.0830 | **0.0162** | 0.0986 | **-2.2268** | **-3.7585** | **-5.1044** |
| | 1 | 6 | **0.0885** | **0.0696** | 0.0174 | **0.0668** | -2.0541 | -3.6834 | -4.8428 |
| GIN-AK$^+$ with D2C | 6 | 1 | 0.0123 | 0.0112 | 0.0150 | 0.0126 | **-2.7513** | **-3.9687** | **-5.1846** |
| | 3 | 2 | 0.0116 | 0.0100 | 0.0168 | 0.0122 | -2.6827 | -3.8407 | -5.1034 |
| | 2 | 3 | 0.0119 | 0.0102 | 0.0146 | 0.0127 | -2.6197 | -3.8745 | -5.1177 |
| | 1 | 6 | 0.0131 | 0.0123 | 0.0174 | 0.0162 | -2.5938 | -3.7978 | -5.0492 |

in regressing graph properties. We hypothesize that D2C-enhanced 1-layer GIN base model is discriminative enough for subgraphs in the dataset, and without expressiveness bottleneck of base model, increasing GNN-AK$^+$'s depth benefits expressiveness, akin to increasing iterations of Subgraph-1-WL. Besides, unlike counting substructure that needs local information within subgraphs, regressing graph properties needs the graph's global information which can only be accessed with increasing GNN-AK$^+$'s (outer) depth.

- Secondly, GNN-AK$^{(+)}$ without D2C suffers from a trade-off between the base model's expressiveness (depth of base model) and the number of GNN-AK$^{(+)}$ layers (outer depth), which is clearly observed in regressing graph properties. We hypothesize that without D2C the 1-layer GIN base model is not discriminative enough for subgraphs in the dataset, and with this bottleneck of base model, GNN-AK$^{(+)}$ cannot benefit from increasing the outer depth. Hence the number of layers of the base model are important for the expressiveness of GNN-AK$^{(+)}$ when D2C is not used.

## A.12 ADDITIONAL RESULTS ON TU DATASETS

We also report additional results on several smaller datasets from TUDataset (Morris et al., 2020a), with their statistics reported in last block of Table 5. The training setting and evaluation procedure follows Xu et al. (2018) exactly, where we perform 10-fold cross-validation and report the average and standard deviation of validation accuracy across the 10 folds within the cross-validation. We take results of existing baselines directly from Bodnar et al. (2021a) with their method labeled as CIN, for references to all baselines see Bodnar et al. (2021a). The result is shown in Table 10.

Table 10: Results on TU Datasets. First section contains methods of graph kernels, second section has GNNs, and third is the method in Bodnar et al. (2021a). The top two are highlighted by **First**, **Second**, **Third**.

| Dataset | MUTAG | PTC | PROTEINS | NCI1 | IMDB-B | RDT-B |
|---------|-------|-----|----------|------|--------|-------|
| RWK | 79.2±2.1 | 55.9±0.3 | 59.6±0.1 | >3 days | N/A | N/A |
| GK ($k = 3$) | 81.4±1.7 | 55.7±0.5 | 71.4±0.31 | 62.5±0.3 | N/A | N/A |
| PK | 76.0±2.7 | 59.5±2.4 | 73.7±0.7 | 82.5±0.5 | N/A | N/A |
| WL kernel | 90.4±5.7 | 59.9±4.3 | 75.0±3.1 | **86.0±1.8** | 73.8±3.9 | 81.0±3.1 |
| DCNN | N/A | N/A | 61.3±1.6 | 56.6±1.0 | 49.1±1.4 | N/A |
| DGCNN | 85.8±1.8 | 58.6±2.5 | 75.5±0.9 | 74.4±0.5 | 70.0±0.9 | N/A |
| IGN | 83.9±13.0 | 58.5±6.9 | 76.6±5.5 | 74.3±2.7 | 72.0±5.5 | N/A |
| GIN | 89.4±5.6 | 64.6±7.0 | 76.2±2.8 | 82.7±1.7 | 75.1±5.1 | **92.4±2.5** |
| PPGNs | 9 0.6±8.7 | 66.2±6.6 | **77.2±4.7** | 83.2±1.1 | 73.0±5.8 | N/A |
| Natural GN | 89.4±1.6 | 66.8±1.7 | 71.7±1.0 | 82.4±1.3 | 73.5±2.0 | N/A |
| GSN | **92.2 ± 7.5** | **68.2 ± 7.2** | 76.6 ± 5.0 | 83.5 ± 2.0 | **77.8 ± 3.3** | N/A |
| SIN | N/A | N/A | 76.4 ± 3.3 | 82.7 ± 2.1 | **75.6 ± 3.2** | 92.2 ± 1.0 |
| CIN | **92.7 ± 6.1** | **68.2 ± 5.6** | **77.0 ± 4.3** | **83.6 ± 1.4** | **75.6 ± 3.7** | **92.4 ± 2.1** |
| GIN-AK$^+$ | **91.3 ± 7.0** | **67.7 ± 8.8** | **77.1 ± 5.7** | **85.0 ± 2.0** | 75.0 ± 4.2 | **94.8 ± 0.8** |

We mark that the performance of GIN-AK$^+$ over IMDB-B is not improved because each graph in the dataset is an egonet, hence all nodes have the same rooted subgraph – the whole graph. The performance of MUTAG and PTC is very unstable, given these datasets are too small: 188 and 344, respectively, and the evaluation method is based on average 10 validation curves over 10 folds.

