# OpenReview forum: "From Stars to Subgraphs: Uplifting Any GNN with Local Structure Awareness"
_ICLR.cc/2022/Conference — ICLR 2022 Poster_

### Official Review · Reviewer_KytV · 2021-10-29

**Correctness:** 1
**Technical Novelty And Significance:** 3
**Empirical Novelty And Significance:** 3
**Recommendation:** 6
**Confidence:** 4

**Main Review:**

**Strengths**
1. Simple approach
2. Good experimental results
3. Good cover of related work

**Weaknesses**
1. Limited selection of benchmark datasets used, e.g., large-scale graphs from OGB and graphlearning.io missing
2. Theoretical statements for 2-WL are misleading. They simply consider the weaker version of the 2-WL, which is known to be equivalent to the 1-WL (see, e.g., https://arxiv.org/abs/2104.14624). This should be clearly stated in the main paper.
3. Proof of Theorem 3 is not sufficient.

**Remark for Theorem 3**
The prove seems to show that there exist pairs of graphs 3-WL cannot distinguish but their architecture can. The proof is not rigorous enough and of a very handwavey nature. **However**, it does not show that their architecture is always at least as powerful. This also has implications for Corollary 3.1.

**Remarks**
1. "Leman" is the preferred spelling, see https://www.iti.zcu.cz/wl2018/pdf/leman.pdf
2. You might also want to cite (Morris et al., 2019) when mentioning the relationship of 1-WL and GNN, see also https://arxiv.org/abs/2104.14624 for a comparison of the results
3. The statement that 1-WL can distinguish almost all graphs needs more context. The result depends on the sizes of the graphs.
4. In  Section 2 you might want to discuss
- https://arxiv.org/abs/2010.01179
- https://arxiv.org/pdf/2101.10320.pdf (also extracting k-hops subgraphs)
5. In Section 3.1 you should use notation for multisets, e.g., {{ ...}}
6. In Section 3.2, you should state the k needed for the results.
7. In Section 4, the description of Emb is clumsy and hard to parse
8. The complexity analysis is not very meaningful. You must derive the cardinality V_U and E_U, which, in general, will depend exponentially on k.



**Summary Of The Paper:**

The paper deals with supervised learning with graphs, specifically graph-level tasks. The paper proposes an algorithm to overcome the expressive limits of the standard GNNs, which are upper-bounded by the 1-WL. The main idea of the paper is to, for each node v, extract the subgraph induced by nodes of at most distance k to node v, and then deploy a GNN on top of each of these subgraphs. Further, the authors study the gains of the expressive power of this architecture compared to the 1-, 2-, 3-WL, and k-WL.

Moreover, the authors propose a subgraph sampling strategy to speed up the computation.

The proposed method is evaluated on large benchmark datasets, mostly stemming from the molecular domain, reporting good performance boosts compared to standard GNN architectures.

**Summary Of The Review:**

The paper proposes a simple and meaningful architecture to enhance the expressive power of most standard GNN architecture.
The experimental results are promising, however, the authors only consider a limited set of benchmark datasets.

However, there are some problems with the theoretical contributions, see above.

---

> ### Author Response · Authors · 2021-11-18
> **Response to Reviewer KytV**
>
> We thank the reviewer for instructive feedback, which helps us revise the submitted manuscript greatly (marked in blue). We additionally address the reviewer’s questions.
>
> >**Limited selection of benchmark datasets used, e.g., large-scale graphs from OGB and graphlearning.io missing**
>
> We have used OGBG-MolHIV (>40,000 graphs) and OGBG-MolPCBA (>400,000 graphs) which are large datasets from OGB --  please see the update dataset statistics table (Appendix A.8 Table 5). Following the reviewer’s suggestion, we conduct additional experiments over 6 datasets from graphlearning.io, which is shown in Appendix A.11. GNN-AK outperforms all baselines in these datasets. We put the number compared with the SOTA method CIN[3] here, for full comparison we suggest the reviewer read Appendix.A.11:
>
> | Method   | MUTAG | PTC| PROTEINS | NCI1|  IMDB-B| RDT-B |
> | ---- | ------ |------ |----- |------ |------ |----|
> |CIN|92.7 ± 6.1 |68.2 ± 5.6 |77.0 ± 4.3 |83.6 ± 1.4| 75.6 ± 3.7| 92.4 ± 2.1|
> |GIN-AK|91.3 ± 7.0|67.8 ± 8.8|77.1 ± 5.7|85.0 ± 2.0|75.0 ± 4.2|94.8 ± 0.8|
>
>
> >**Theoretical statements for 2-WL are misleading. They simply consider the weaker version of the 2-WL, which is known to be equivalent to the 1-WL**
>
> We have updated the reference to the connection between 1-WL and 2-WL in page 5 footnote. We thank the reviewer for pointing it out.
>
> >**Proof of Theorem 3 is not sufficient. The proof seems to show that there exist pairs of graphs 3-WL cannot distinguish but their architecture can. The proof is not rigorous enough and of a very handwavey nature.**
>
> We have updated the proof of Theorem 3 in Appendix A.4 to make it more formal. We state that the definition of “A is no less powerful than B” here: we call A is no less powerful than B if there exists a pair of non-isomorphic graphs that cannot be distinguished by B but can be distinguished by A. This definition follows several papers: Zhengdao et al. 2020 [2], and Bodnar et al. 2021 [3].
>
> >**There are some problems with the theoretical contributions.**
>
> We additionally updated proofs for all theorems and conjectures to make them formal: updated theorem 4 proof, and provided the proof for conjecture 1. We hope this can clear any misunderstandings regarding the accuracy and coverage of the proofs.
>
> >**Remarks**
>
> 1.We now use “Leman” in our paper.
> 2.We cited (Morris et al., 2019) in Page 4.
> 3.We add context for “1-WL can distinguish almost all graphs” in Page 2.
> 4.We have added the mentioned two literatures into Related Work.
> 5.In section 3.1 we marked notation for multisets, following the GIN paper [1].
> 6.In section 3.2 we stated k in Theorem 4, for others k is not necessary.
> 7.In section 4, we revised the description of Emb.
>
>
> >**The complexity analysis is not very meaningful. You must derive the cardinality V_U and E_U, which, in general, will depend exponentially on k.**
>
> We agree that the cardinalities  of  $V_U$ and $E_U$ should depend on $k$, however it is non-trivial to derive without assumption on the graph structure, and the complexity analysis in our paper is mainly given to identify the memory bottleneck. The current analysis helped this and inspired the designing of SubgraphDrop, resulting in GNN-AK-S with complexity independent of $k$ (equivalent to roughly $R$ times the overhead of base model).
>
> **Reference**:
>
> [1] Xu, Keyulu, et al. 2018 "How powerful are graph neural networks?."
>
> [2] Zhengdao et al. 2020  "Can graph neural networks count substructures?."
>
> [3] Bodnar, Cristian, et al. 2021 "Weisfeiler and lehman go cellular: Cw networks."
>
> ---------------------
> We thank the reviewer once more for their detailed feedback, which helps us to revise the paper greatly. We kindly ask the reviewer to consider raising their score if their concerns were appropriately addressed.

---

> > ### Comment · Reviewer_KytV · 2021-11-22
> > **Response**
> >
> > The authors addressed most of my mentioned issues satisfactorily, _excluding_ the running time analysis. I increased my score slightly.
> >
> > They should also discuss the two mentioned papers also incorporating $k$-hops subgraphs in the related work section.

---

> > > ### Author Response · Authors · 2021-11-23
> > > **Thank you**
> > >
> > > We thank the reviewer for satisfying our revision.
> > >
> > > In the first revision we mistakenly forgot the two papers mentioned by the reviewer, and now they are included. Please see page 2 for the RNI paper and page 3 for the identity-aware paper.

---

### Official Review · Reviewer_LzXw · 2021-11-01

**Correctness:** 3
**Technical Novelty And Significance:** 3
**Empirical Novelty And Significance:** 3
**Recommendation:** 6
**Confidence:** 5

**Main Review:**

Strengths:
1. The proposed GNN-AK framework is novel for uplifting the expressiveness of GNN. It is an interesting idea to use GNN as a kernel on the induced subgraph. The design for both the GNN-AK layer and subgraph sampling are reasonable.
2. The proofs of theorems appear to be correct.
3. This paper provides a comprehensive experimental study, including both qualitative analysis and quantitative results, to show the effectiveness of GNN-AK framework. The ablation study shows that each component in the framework works well.

Weaknesses:
1. This paper didn’t explain how it gets d_{i|j}^{(l)} from distance-to-centroid. The Sigmoid function was used to combine the D2C feature.  It is counter-intuitive if a node with a larger D2C gets a higher weight during aggregation. It would be better to provide more details on this.
2. The dataset statistics are missing.
3. What would be the performance of the proposed method when a different way of combining the centroid, subgraph, and context encoding is used?
4. Some recent SOTA methods are missing:
Bodnar, Cristian, et al. "Weisfeiler and lehman go cellular: Cw networks." arXiv preprint arXiv:2106.12575 (2021).
Ying, Chengxuan, et al. "Do Transformers Really Perform Bad for Graph Representation?." arXiv preprint arXiv:2106.05234 (2021).


**Summary Of The Paper:**

The paper proposes GNN-AK, a framework that can use GNN as a kernel to encode local features. It generalizes the local neighbour aggregation in Massage Passing Neural Networks (MPNN) from a star-like pattern to a more flexibly defined subgraph. The paper also provides theoretical support that shows the superiority of the proposed method to 1&2-WL in terms of expressiveness.  The experimental results demonstrate that the proposed method outperforms other SOTA baselines on 7 different datasets.

**Summary Of The Review:**

This paper puts forward a novel framework that can easily plug different kinds of GNN and improve the expressiveness. The proofs and experimental results well support the proposed method. Some concerns are listed above.

---

> ### Author Response · Authors · 2021-11-18
> **Response to Reviewer LzXw**
>
> We thank the reviewer for the helpful review, and address the reviewer’s question as follows. We also updated the paper dramatically, marked in blue.
>
>
> >**This paper didn’t explain how it gets d_{i|j}^{(l)} from distance-to-centroid. The Sigmoid function was used to combine the D2C feature. It is counter-intuitive if a node with a larger D2C gets a higher weight during aggregation. It would be better to provide more details on this.**
>
> We record the distance to centroid value for every node in every subgraph, and the value is categorical instead of continuous. We then encode the categorical D2C feature to embedding through an embedding layer at each $l$-th layer (nn.Embedding in PyTorch), resulting in $d^{(l)}_{i|j}$. Because the D2C is a categorical feature at the beginning, it doesn’t have the problem mentioned by the reviewer -- nodes with different D2C have different embedding vectors which are provided as a form of context, but not explicitly higher or lower scalar weights.
>
> >**The dataset statistics are missing.**
>
> We have included the dataset table in Appendix A.8 Table 5. Notice that the table includes additional six datasets from TUDataset, with their result shown in Appendix A.11.
>
> >**What would be the performance of the proposed method when a different way of combining the centroid, subgraph, and context encoding is used?**
>
> We have conducted an ablation study with removing just 1 encoding in Appendix A.10. Table 7.
> For combining these encodings using another way, we have tested replacing concatenation to sum during the model development stage (however we didn’t save the result down), and the result is very similar, but slightly worse than concatenation.
>
> >**Some recent SOTA methods are missing: Bodnar, Cristian, et al. "Weisfeiler and lehman go cellular: Cw networks." arXiv preprint arXiv:2106.12575 (2021). Ying, Chengxuan, et al. "Do Transformers Really Perform Bad for Graph Representation?." arXiv preprint arXiv:2106.05234 (2021).**
>
> We have updated the related work section to include the reviewer’s suggestions: these two papers are now cited. Also we have added CIN as baseline (Cw network) to Table 3. And conducted additional experiments on 6 datasets from TUDataset in Appendix A.11 to compare with CIN.
> We put the number compared with the SOTA method CIN[3] here, for full comparison we suggest the reviewer read Appendix.A.11:
>
> | Method   | MUTAG | PTC| PROTEINS | NCI1|  IMDB-B| RDT-B |
> | ---- | ------ |------ |----- |------ |------ |----|
> |CIN|92.7 ± 6.1 |68.2 ± 5.6 |77.0 ± 4.3 |83.6 ± 1.4| 75.6 ± 3.7| 92.4 ± 2.1|
> |GIN-AK|91.3 ± 7.0|67.8 ± 8.8|77.1 ± 5.7|85.0 ± 2.0|75.0 ± 4.2|94.8 ± 0.8|
>
> Notice that we haven’t added Graphormer from the second mentioned paper to these tables as their result is not comparable: Graphnormer is a pretrained model that uses additional datasets.
>
> --------
> We thank the reviewer again for supporting our work, and we kindly ask the reviewer to consider raising the score if we fully addressed the reviewer’s concern.

---

> > ### Author Response · Authors · 2021-11-28
> > **Looking forward to your feedback**
> >
> > As the discussion period is ending in 2 days, we would like to ask if there is still any issue that we can address? We are looking forward to your response.

---

> > > ### Comment · Reviewer_LzXw · 2021-11-28
> > > **Thanks for the response**
> > >
> > > All my concerns are addressed. Please add in the clarification of Point 1 in the revised version. I prefer keeping the current score.

---

### Official Review · Reviewer_oHnh · 2021-11-02

**Correctness:** 4
**Technical Novelty And Significance:** 3
**Empirical Novelty And Significance:** 3
**Recommendation:** 8
**Confidence:** 4

**Main Review:**

# Strengths

- The motivation behind the proposed method is very clearly described in the introduction.
- The high-level insights described at the beginning of Section 3 give a good overview of the paper before describing the details.
- The proposed subgraph 1-WL procedures are a nice tool for analyzing the expressive power of the proposed models. In particular, I like the idea of defining an isomorphism test that relies on WL as a subroutine (i.e. the 1-Subgraph-1-WL*) and thus "uplifting" the expressive power of WL.
 - The theoretical analysis performed on top of these newly proposed isomorphism tests contains useful results. Although some of the results are relatively obvious, they help provide a description of the capabilities of the proposed framework.
- In particular, I think Theorem 4 is the most original since it manages to establish a connection with all the tests in the k-WL hierarchy for $k \geq 3$.
- The complexity analysis section is compact and clear. Like other recent papers, the proposed method manages to benefit from cutting through the k-WL hierarchy by achieving more advantageous complexity tradeoffs compared to k-WL based approaches.
- I like that the authors have taken a careful approach with the subgraph sampling rather than simply implementing a naive sampling approach. The proposed sampling takes into account how the nodes are covered by different subgraphs and the method itself is adapted to avoid potential complications at testing time when no sampling is used.
- The experiment in Table 1 is very nice and clearly demonstrates the claimed capacity of the framework to "uplift" existing GNNs. It is also encouraging that the framework brings improvements irrespective of the base GNN that was tried.
- The results for the real-world experiments are good. Although, some stronger baselines could have been included (see the next section)
- The ablation study in Table 4 provides a useful overview of the tradeoff between computational expenses and performance.

# Weaknesses

- There seems to be a discrepancy between the high-level view of the GNN-AK model provided in Equation (2) and the actual implementation in Equation (6). The context representation from Equation (6) is based on subgraphs rooted at neighbours of node v and therefore it might be based on information that is outside the subgraph of v, thus becoming misaligned with Eq(2) and the proposed subgraph WL equations.
- The subgraph sampling strategy is often referred to as a sort of Dropout for GNNs. However, the evidence that this subgraph sampling strategy improves generalisation in any way is weak. Table 4, which is studying this does not provide any standard errors around the provided scores. This is particularly important as more subsampling induces more noise into these measurements. It is therefore very difficult to assess the statistical significance of these scores. Even ignoring the lack of error bars, on ZINC no level of subgraph sampling produces improvements in the reported MAE compared to the full GIN-AK and in the case of CIFAR-10, the differences are far from being scientifically significant.
- While I understand the positives that subgraph sampling brings, training-time computationally improvements are generally not so useful. Ideally, one would like to move as much of the processing from inference time to training time because that is the main setting where latency and computational constraints start to matter. However, the model does need to use all the subgraphs at inference time.
- A stronger baseline for ZINC and MOLHIV exists: *Weisfeiler and Lehman Go Cellular: CW Networks (NeurIPS 2021)* (https://arxiv.org/abs/2106.12575). In fact, this work is also particularly relevant for this paper since it is also relying on induced subgraphs for "uplifting" existing GNNs, but it does so in a different way.

# Minor weaknesses

- In the introduction (last paragraph of page 2), the paper claims that sufficient conditions are derived under which GNN-AK can distinguish two graphs. This is misleading because these sufficiency conditions are only conjectured on page 1. While some empirical evidence is provided in favour of this conjecture, it does not represent a definitive result.
- It is unclear to me why Conjecture 1 is a conjecture and not an actual result. On a first look, it seems rather immediate that the assumptions of the Conjecture imply distinguishability (which doesn't mean that is indeed the case). What I am trying to say is that there should be at least a brief description of the difficulties surrounding the proof of the result and what the main impediments are.
- Nit: In Table 2 it is not immediately clear which numbers correspond to the outer vs inner loop. I suggest reformatting the table to make this more clear.

**Summary Of The Paper:**

The paper introduces a framework that can extend the expressive power and empirical performance of a base GNN by using an induced subgraph-based local aggregation method that substitutes the conventional multiset aggregation GNNs typically perform. The authors provide theoretical analysis for this framework, study its scalability for practical applications, and presents good empirical results.

**Summary Of The Review:**

The strengths of the paper outlined above generally outweigh the weaknesses, so I am therefore recommending the paper for weak acceptance for now. I am looking forward to the discussion period and I am willing to change my score once my points are addressed.

EDIT: I have raised my score to 8.

---

> ### Author Response · Authors · 2021-11-18
> **Response 1/2**
>
> We thank the reviewer for suggestive feedback and we have updated our paper accordingly, marked in blue. We address the reviewer’s question one by one.
>
> >**There seems to be a discrepancy between the high-level view of the GNN-AK model provided in Equation (2) and the actual implementation in Equation (6).**
>
> Indeed, our novel concrete realization has 3 types of encodings. Firstly, our theory supports the usage of subgraph encoding and centroid encoding (Subgraph-1-WL also supports using centroid encoding, potentially making *HASH* over subgraphs more discriminative), but *does not* refute the usage of context encoding. In fact, using context encoding further improves the empirical expressiveness of GNN-AK, while theoretically analyzing its expressiveness is non-trivial in our Subgraph-1-WL paradigm, and thus we leave it to future work. As in most deep learning work, practical design often precedes theoretical understanding: likewise, GNN-AK is more empirically expressive *with* than  *without* context encoding, where the latter is fully supported by our theory through Subgraph-1-WL. To fully address the reviewer’s concern we evaluate GNN-AK without context encoding on all simulation datasets and two real-world datasets in appendix A.10 Table 8, where we observe that GNN-AK outperforms GNN-AK without context encoding, while the latter still significantly improves over base GNN. We also note that theory shouldn’t block designing better models: MLP is technically a universal approximator, but new models like CNN and transformer are designed to achieve better practical performance. Moreover, expressiveness is not always correlated with real-world performance (in fact, 1-WL already can distinguish many graphs in most  datasets, see Figure 3 of https://arxiv.org/pdf/2105.05911.pdf), while the trade-off between generalization and expressiveness plays an important role and is highly affected by designs.
>
> To help understand the actual reason for the superior performance of GNN-AK, Appendix A.10 Table 8 also shows additional ablation studies to decouple all designs and identify the root of the improvements. As we can see, the theoretically substantiated designs (GIN-AK w/o Ctx&D2C) indeed provide the largest improvement (42.3% error reduction on ZINC-12K, 7.69% absolute acc improvement on CIFAR10) over the base model, while context encoding and D2C provides additional benefit (additional  8.5% error reduction on ZINC-12K, additional 4.65% absolute acc improvement on CIFAR10). As we noticed that GIN-AK w/o Ctx&D2C offers comparatively less improvements on simulation datasets, we hypothesize that this is because of weak expressiveness of 1-layer base GIN when D2C is not used, and increasing base model’s depth is important. Hence we further studied the impact of the base model’s depth under the theory of Subgraph 1-WL in Table 9, and showed that with a deeper base model, GIN-AK w/o Ctx&D2C still provides a significantly large improvement over the base model. All observations in Table 9 are fully supported by proposition 1, we suggest the reviewer to read the **extensively updated Appendix A.10**.
>
> To summarize, our theory justified the significant performance improvement of GIN-AK without context encoding by Subgraph-1-WL, and the performance of GIN-AK without both D2C and context encoding by Subgraph-1-WL^*. Proposition 1 justified the empirical study of base model’s depth with and without D2C. Context encoding and D2C are important designs to benefit GNN-AK even further, which support the novelty and importance of our concrete realization. In future work we plan to theoretically prove the expressiveness benefit of context encoding and D2C within GNN-AK.
>
> >**The subgraph sampling strategy is often referred to as a sort of Dropout for GNNs. However, the evidence that this subgraph sampling strategy improves generalisation in any way is weak.**
>
> We agree with the reviewer that the generalization argument is too weak which is originally motivated by Dropout that benefits generalization. As we don’t have enough evidence to support this argument, we removed it from our paper to keep it as future work. Nevertheless we do want to mention that there may exist other potential benefits of SubgraphDrop.
>
> The reviewer also wants to know the standard deviation of the Table 4, we include the result here.
>
> | Dataset   | R=1 | R=2| R=3| R=4| R=5|
> | ----------- | ------ |------ |----- |------ |------ |
> | ZINC-12K  | 0.1216 ± 0.0057 |0.0929 ± 0.0025|0.0846 ± 0.0019|0.0852 ± 0.0040|0.0854 ± 0.0023|
> | CIFAR10   | 71.68 ± 0.43 |72.07 ± 0.40|72.39 ± 0.38|72.20 ± 0.35|72.32 ± 0.39|

---

> > ### Author Response · Authors · 2021-11-18
> > **Response 2/2**
> >
> > >**While I understand the positives that subgraph sampling brings, training-time computationally improvements are generally not so useful. Ideally, one would like to move as much of the processing from inference time to training time**
> >
> > Both training time and inference time are important, but in different perspectives: training time is super important during the model development stage, while inference time is important during the model deployment stage. As training set is typically greatly (100 times) larger than validation and test set, and is passed to model thousands of times (the number of training epochs) with backpropagation (inference is a lot faster than training as it does not need backpropagation), improving training efficiency can greatly reduce training time and faster model development. This is especially true for very large graphs, often dealt with in practical scenarios -- several popular works have primarily focused on improving training efficiency in GNNs for this reason [1,2,3,4]. Moreover, training efficiency offers multi-fold benefits in massively parallel scenarios like hyperparameter tuning (where many training jobs need to be run, needing tremendous computational resources). What’s more, the SubgraphDrop mainly solves the memory bottleneck problem: without it, one cannot successfully train the model in a single GPU for larger datasets, as batch size cannot be too small with BatchNorm used. For reducing inference time during the model deployment stage, there are other options including distillation and quantization, which is associated with a separate line of work.
> >
> > >**A stronger baseline for ZINC and MOLHIV exists: Weisfeiler and Lehman Go Cellular: CW Networks (NeurIPS 2021)**
> >
> > We have cited the paper in section 2, and compared with the paper (CIN) in Table 3. Even more, we have added 6 additional datasets from TUDataset and compared with CIN in Appendix A.11, where GIN-AK outperforms CIN significantly even without any hyperparameter tuning. Last but not least, CIN has to solve the subgraph isomorphism test problem during the preprocessing stage which in general is not scalable, while our preprocessing is linear in the number of nodes.
> >
> > We put the number compared with the SOTA method CIN[3] here, for full comparison we suggest the reviewer read Appendix.A.11:
> >
> > | Method   | MUTAG | PTC| PROTEINS | NCI1|  IMDB-B| RDT-B |
> > | ---- | ------ |------ |----- |------ |------ |----|
> > |CIN|92.7 ± 6.1 |68.2 ± 5.6 |77.0 ± 4.3 |83.6 ± 1.4| 75.6 ± 3.7| 92.4 ± 2.1|
> > |GIN-AK|95.7 ± 3.3| 71.5 ± 5.2| 78.5 ± 3.0| 85.2 ± 1.1 |75.8 ± 4.2| 95.1 ± 1.4|
> >
> >
> > >**The paper claims that sufficient conditions are derived under which GNN-AK can distinguish two graphs. This is misleading because these sufficiency conditions are only conjectured on page 1.**
> >
> > We change Conjecture 1 to Proposition 1, and give a formal proof in Appendix A.6. We prove it by contradiction.
> >
> > >**In Table 2 it is not immediately clear which numbers correspond to the outer vs inner loop.**
> >
> > Thanks for this suggestion -- we have updated Table 2 accordingly for clarity.
> >
> > **Reference**:
> >
> > [1] GraphSaint: Graph Sampling Based Inductive Learning Method (Zeng et al, ICLR’20)
> >
> > [2] ClusterGCN: An Efficient Algorithm for Training Deep and Large Graph Convolutional Networks (Chiang et al, KDD’19)
> >
> > [3] Redundancy Free Computation Graphs for Graph Neural Networks (Jia et al, KDD’20)
> >
> > [4] Global Neighbor Sampling for Mixed CPU-GPU Training on Giant Graphs (Dong et al, KDD’21)
> >
> > ---------
> > We thank the reviewer once more for their detailed feedback, which helps us to make a strong revision. We kindly ask the reviewer to consider raising their score if their concerns were appropriately addressed.

---

> > > ### Comment · Reviewer_oHnh · 2021-11-21
> > > **Response to authors**
> > >
> > > Thank you for your response! The ablation study you added is very useful and the other changes are also a step in the right direction.
> > >
> > > Before deciding on my final score, I have a few more comments and clarifying questions:
> > >
> > > **Re Discrepancy between Equation (2) and the actual implementation in Equation (6).**
> > >
> > > I think there was some misunderstanding about this point. I did not mean that the authors should NOT have considered such design choices in practice, but rather that:
> > >
> > > - At the current stage of the manuscript the presentation around these aspects is very misleading. The proposed model is described as a "concrete realization" of GNN-AK, when in fact the model using the context is outside the GNN-AK framework introduced earlier.
> > > - This creates a false expectation that the model used in the experiments has the same expressive power as GNN-AK when in fact it is at least as powerful as GNN-AK.
> > > - Finally, this raises the question: If the version with context is what you use in practice, why not include the context in Equation (2) and study the expressive power of this more general model with and without context?
> > >
> > > **TU Datasets Experiments**
> > >
> > > Which cross-validation splits did you use for these experiments? From my experience, it is quite easy to get better results on these datasets by using a different set of splits than those used by the baselines.

---

> > > > ### Author Response · Authors · 2021-11-23
> > > > **Response to Reviewer oHnh**
> > > >
> > > > Thank you for encouraging our revision and providing additional feedback!
> > > >
> > > > >**Discrepancy between Equation (2) and the actual implementation in Equation (6).**
> > > >
> > > > Based on the reviewer's suggestion, we have rewritten the section 4 to present two variants of model: GNN-AK and GNN-AK+, where the first one is based on theory presented in section 3 and the later is with additional context encoding and D2C. We hope this can avoid the clarity issue mentioned by the reviewer. Please take a look at the new revision and let us know whether this helps. Accordingly, in experiment section we present both GNN-AK and GNN-AK+, while GNN-AK is only shown with GIN base model to save space. As we are still running these result for GIN-AK, we leave some part blank in the table for now but we will include these result in the final paper.
> > > >
> > > > >**TU Datasets Experiments**
> > > >
> > > > We were using random split previously, and we are now working on the specific split the reviewer mentioned. We will update the result with response once finished. Thank you for pointing this out.

---

> > > > > ### Author Response · Authors · 2021-11-26
> > > > > **Updated Result on TUDataset**
> > > > >
> > > > > We have got the updated result for TUDataset, we use the same split, same configuration and same evaluation as CIN. Please see the following table:
> > > > >
> > > > > | Method   | MUTAG | PTC| PROTEINS | NCI1|  IMDB-B| RDT-B |
> > > > > | ---- | ------ |------ |----- |------ |------ |----|
> > > > > |CIN|92.7 ± 6.1 |68.2 ± 5.6 |77.0 ± 4.3 |83.6 ± 1.4| 75.6 ± 3.7| 92.4 ± 2.1|
> > > > > |GIN-AK|91.3 ± 7.0|67.8 ± 8.8|77.1 ± 5.7|85.0 ± 2.0|75.0 ± 4.2|94.8 ± 0.8|
> > > > >
> > > > > The performance of GIN-AK over IMDB-B is not improved because each graph in the dataset is an egonet, and hence all nodes have the same rooted subgraph -- the whole graph.
> > > > >
> > > > > The performance of MUTAG and PTC is very unstable, given these datasets are too small: 188 and 344, respectively, and the evaluation method is based on average 10 validation *curves* over 10 folds.
> > > > >
> > > > > We will update these results to the main paper in final stage. We hope we have addressed the reviewers' questions.

---

> > > > > ### Comment · Reviewer_oHnh · 2021-11-27
> > > > > **Response to authors (second round)**
> > > > >
> > > > > **Re latest changes**
> > > > >
> > > > > Thank you for addressing the discrepancy between the equations! I believe things are much clearer now in this respect and having both versions evaluated is a big plus.
> > > > >
> > > > > I am also glad you have re-run the TU Dataset experiments with the GIN paper splits! While the TU results do not influence my opinion of the paper (they are a very noisy dataset with well-known limitations), it is obviously important to use the same splits as the baselines to avoid misleading the readers.
> > > > >
> > > > > **On similar concurrent work**
> > > > >
> > > > > Overall, I think one of the main concerns of the other reviewers is that the core idea of the paper surfaced in a couple of other papers. Indeed, it is a striking coincidence to see so many concurrent works on this topic:
> > > > >
> > > > > - Two other papers also under review at ICLR essentially explore similar ideas: https://openreview.net/forum?id=uxgg9o7bI_3 and https://openreview.net/pdf?id=dFbKQaRk15w
> > > > > - There is a concurrent NeurIPS paper (https://arxiv.org/abs/2110.13197) that has only been public (on arxiv) since October
> > > > > - Another concurrent NeurIPS work focusing more on subgraphs and the reconstruction conjecture: https://arxiv.org/abs/2110.00577, which has also been made public only very recently.
> > > > > - Ego GNNs (https://arxiv.org/abs/2107.10957?context=cs) which have been public since July
> > > > >
> > > > > However, these can all be considered concurrent and (IMO) should not influence the reviewers' opinion about the paper. And if they do, it should be perhaps in a positive way since this paper manages to obtain favourable results compared to many of these while also being well executed from a theoretical point of view.
> > > > >
> > > > > **Score update**
> > > > >
> > > > > As I explained above, I believe the work should be judged as being novel despite these concurrent works. Taking everything together, I believe my concerns about the paper have been addressed and I will raise my score to accept (8).

---

> > > > > > ### Author Response · Authors · 2021-11-28
> > > > > > **Thank you**
> > > > > >
> > > > > > We thank you sincerely for your positive feedback and the effort on helping us revising the paper!
> > > > > >
> > > > > > 1. Regarding TUDataset, although we have successfully followed the same configuration as previous literature, we have to admit that the evaluation procedure on these datasets seems problematic: the average of validation curve on 10 folds requires the validation curve being aligned in all folds in order to achieve good performance, which is impossible for these small datasets. We wish better evaluation can be designed for these small datasets in future.
> > > > > >
> > > > > > 2. Regarding concurrent work. Indeed, there are so many papers presented in the same period using the subgraph idea, and the two submissions of ICLR 2022 mentioned by the reviewer are similar to ours in terms of key idea, but different in designs. Also we would like to emphasize the contribution of SubgraphDrop here: all subgraph based methods will have the memory and runtime overhead issues we mentioned, and we believe the design of SubgraphDrop is important for using these methods in real-world. Finally, we will summarize all these works in the final version of the paper to help future researchers.
> > > > > >
> > > > > > We will open source our code soon! We thank the reviewer's effort on the paper again, and appreciate the positive feedback sincerely.

---

### Official Review · Reviewer_MkR2 · 2021-11-02

**Correctness:** 3
**Technical Novelty And Significance:** 2
**Empirical Novelty And Significance:** 3
**Recommendation:** 6
**Confidence:** 5

**Main Review:**

**Strengths**:
-	The paper is nicely written, has a good flow and is easy to follow.
-	The idea of ego-nets is simple and straightforward to implement. Contrary to other GNNs that improve expressivity using some kind of structural property (e.g., Bouritsas et al., arxiv’20,  Thiede et al., NeurIPS’21, Barceló et al., NeurIPS’21 and others), GNN-AK requires fewer hyperparameters to be selected by the user (mainly the size $k$ of the ego-net).
-	Practical solutions that improve scalability without sacrificing performance are proposed.
-	The experimental section is extensive and the method obtains competitive results in many datasets, which is a good indication that the proposed architecture will work well in practice.

**Weaknesses**:
My main concerns about this paper have to do with (1) its novelty, (2) the fact that the practical instantiation contains multiple choices that are not justified from the theory and obfuscate the actual underlying reason for the good empirical performance. In more detail:

1. **Novelty**: The idea of using ego-nets to improve the expressive power of GNNs is not completely new. As the authors also mention, this idea has been used several times in the past. For example, a reference that the authors have missed, “Identity-aware Graph Neural Networks”, You et al., AAAI’21, is also closely related and it goes one step further in terms of the theoretical results, showing that by identifying the root note in each ego-net (implicitly present in this work – via the distance to centroid), one can also count cycles. Other examples are e.g., k-hop GNNs, Nikolentzos et al., ’20 with also theoretical results orthogonal to this work (showing that their GNN can regress certain graph properties)  and Ego-GNNs, Sandfelder et al.’20. Although there are differences between the above, the main idea is similar, therefore I would like to ask the authors to explain what makes their method significantly different. This brings me to my second concern.
2. **Architecture instantiation**:
    -   The current arguments the authors used to differentiate their work from other related mainly boil down to architecture/implementation details (e.g., they write “unlike GNN-AK, they only propose to use a final root node embedding” and “can be viewed as a special case of GNN-AK, which only computes a context embedding (one of three types of embeddings GNN-AK uses)”), but their theoretical results do not seem to advocate for the necessity of these architectural choices.
    -  Further, more ad hoc choices that do not seem to result from the theory are presented later in the paper (3 types of embeddings, gating mechanism with the distance to centroid, etc.). This begs the following question: What is the actual reason for the arguably good empirical results? Is it due to the improved expressivity or due to engineering?
    -	Also, note that some of the choices of the authors (random-walks to extract the subgraphs, dropout on the ego-nets) introduce randomness in the algorithm which (1) sacrifices permutation equivariance (which is not necessarily bad), but (2) improves expressivity (e.g., see related works that use random features or other randomized algorithms for GNNs), which might be also a confounding factor.

3. **Theory**:
   - The theoretical results up to Corollary 3.1 are similar to previous work (e.g., Bouritsas et al., arxiv’20, Bodnar, Frasca, Wang et al., ICML'21, Bodnar, Frasca et al., NeurIPS'21 and others). Unfortunately, the bar has been set higher in research on GNN expressivity, since currently, most architectures that are proposed enjoy these properties (more expressive than 1-WL, no less expressive than 3-WL – shown by counterexample), hence in order for the theoretical evidence to be convincing, in my opinion, further arguments are required (e.g., substructure counting, or graph property prediction - possibility/impossibility results).
   - The authors do provide a novel theoretical argument in Theorem 4, which is good, however, its practical consequences are not clear to me (I don’t understand the following sentence: “This opens a future direction of generalizing rooted subgraph to general subgraph (as in k-WL) while keeping number of subgraphs in $O(|\mathcal{V}|)$.”).
   - More clarity is needed in Conjecture 1 with respect to the magnitude of $k$. If it is $O(n)$, then this resembles the reconstruction conjecture (Kelly, 1957), if it is $O(1)$, then this would imply a polynomial-time solution to isomorphism, hence it is unlikely to hold.
   - Can you theoretically compare ego-net enabled GNNs with other modern GNNs, such as the ones that use substructures or other structural properties to improve expressivity? Note that these methods can also distinguish 3-WL failed graphs, without necessarily resorting to an algorithm of higher computational complexity during training, which is the condition of your Theorem 3.

**Other comments**:
- **Experiments**:
   - What version of GNN-AK do you use in Tables 1 and 2? Is it the vanilla GNN-AK or it contains the instantiation details mentioned in the previous sections?
   - Minor: I suspect that this method might suffer from scalability issues when dealing with denser graphs than those you tested on (e.g., social networks) since its complexity might grow with $O(n^2)$. Have the authors considered testing on this domain?
- The following claim in the intro should be relaxed:  “We also give sufficient conditions under which GNN-AK can successfully distinguish two non-isomorphic graphs.”. As far as I understood, this is based on a conjecture rather than a proven fact.
-  Theorem 4: (1) Notation clash ($k$ is now used for k-WL and $d$ for the num of hops of the ego-net) and (2) there are clarity issues in the proof.  I am not sure I follow the claim “and the rewiring of constructing its non-isomorphic counter graph has picked two edges that cannot be included by any k-hop ego-nets with k ≤ 4”. Is that a known fact from the original CFI paper? Maybe the authors would like to expand their explanations. Does this proof imply that with $k\geq4$ you might be able to distinguish two CFI graphs?

### --------------------- After rebuttal ---------------------
I have increased my score slightly, this time voting for weak acceptance. Please see my final comment for a justification, as well as the discussion with the authors for a detailed explanation of the remaining concerns.




**Summary Of The Paper:**

This manuscript presents a general-purpose technique to improve GNN expressivity, dubbed as GNN-AK, that replaces the conventional neighbourhood/star-like aggregation (multi-set of neighbour embeddings) with an ego-net level aggregation. In particular, at each layer of GNN-AK, the authors first extract all $|\mathcal{V}|$ ego-nets of the original graph, then they apply a GNN on each one of them, and finally, they collect their outputs into node-wise representations.  The expressive power of different variants of their model is theoretically analysed (more expressive than 1-WL when the ego-net aggregation is as expressive as 1-WL, and no less powerful than 3-WL, when the ego-net aggregation is as expressive as 3-WL) and a series of design choices for a practical instantiation is provided. The authors extensively evaluate their method on both synthetic and real-world datasets and ablate the influence of some of the moving parts in the overall performance.

**Summary Of The Review:**

The main idea in this paper is simple, easy to implement and the empirical results are good. However, (1) this very idea is not that novel since ego-nets have been used in the past,  (2) most theoretical results are similar to prior work and do not make a particularly compelling case, while other works on ego-nets have gone further in that respect and (3) the multitude of engineering choices obscures the root causes for the improved empirical performance. Therefore, I am leaning towards rejection, but I would encourage the authors to clarify their position w.r.t. the above and make their statements more clear since it seems that ego-nets have good potential in terms of practical performance

---

> ### Author Response · Authors · 2021-11-18
> **Response to novelty questions**
>
> We thank the reviewer for the thorough and detailed review on our submission. We have revised our submission significantly based on the reviewer’s questions and suggestions, marked in blue. We respond to the reviewers’ concerns and questions one by one.
>
> >**Novelty: ... I would like to ask the authors to explain what makes their method significantly different.**
>
> Our work is the first **general** framework of uplifting GNN with **theoretical** support, great **scalability**, and remarkable **practical** performance. We thoroughly discussed the difference between our work and all existing literature one by one, and **updated section 2** in the paper. We also want to note that based on ICLR 2022 policy, all papers after June 5, 2021 are contemporaneous to our paper. Nevertheless, we are happy to discuss and summarize these new papers.
>
> 20-[Bouritsas et al., arxiv’20] Improving Graph Neural Network Expressivity via Subgraph Isomorphism Counting
> 21-[Barceló et al., NeurIPS’21] Graph Neural Networks with Local Graph Parameters **[after June 5, 2021]**
>
> These two papers compute subgraph based features to augment the input. The two papers are significantly different from our work: the designed subgraph features are handcrafted, while our work directly learns representations from subgraphs.
>
> 21-[Thiede et al. NeurIPS 21] Autobahn: Automorphism-based graph neural nets **[after June 5, 2021]**
> 21-[Bodnar et al. NeurIPS 21] Weisfeiler and Lehman Go Cellular: CW Networks **[after June 5, 2021]**
>
> These two papers share some similarities. They design GNN to use certain subgraph patterns (cycles and paths, to be specific) in the message passing phase. As mentioned by the reviewer, these papers have lots of hyperparameters and designs for choosing subgraph patterns. Moreover,  their preprocessing requires solving the NP-hard subgraph isomorphism problem (see footnote 1 on page 5 of Thiede et al.), which is not scalable compared  to our $O(k|E|)$ preprocessing complexity.
>
> 21-[You et al. AAAI 21] Identity-aware Graph Neural Networks
>
> Similar to our work, ID-GNN also uses k-egonet --  however, their goal is making each node recognize itself during message passing, where our design is theoretically motivated from the proposed Subgraph-1-WL (Defn 3.1).  As WL has multiple iterations to refine the color, Subgraph-1-WL also has multiple iterations, resulting in GNN-AK having multiple (outer) layers. However ID-GNN only has *one* outer layer: it applies a multi-layer base GNN over k-egonet once. ID-GNN limits the receptive field of each node within the k-egonet where our GNN-AK can easily gain information outside the k-egonet by stacking multiple GNN-AK layers. From a design perspective, the ID-GNN only uses centroid encoding mentioned in our paper. What’s more, directly using k-egonet introduces scalability overhead which is not addressed in the ID-GNN paper, while we designed SubgraphDrop to greatly improve the framework’s usability.
>
> 20-[Nikolentzos et al.’20] k-hop graph neural networks
> 21-[Sandfelder et al.’21] Ego-GNNs **[after June 5, 2021]**
>
> K-hop GNN uses k-egonet in a specially designed way: it encodes a rooted subgraph via sequentially passing messages from higher hops in the subgraph to lower hops until it reaches the rooted node, and then use the rooted node as encoding of the subgraph. Ego-GNN is just a limited special case of GNN-AK, with SGC as base model and only uses contextual encoding. They do share similarity with GNN-AK, however the k-hop GNN and Ego-GNN 1) are not a general framework to uplift any existing GNNs; 2) do not connect with Subgraph-1-WL; 3) do not go beyond subgraph encoding; 4) do not address scalability overhead.
>
>
> **To summarize our novelties:**
>
> 1. **Theoretical novelty**: we are the first to propose Subgraph-1-WL, analyze its expressiveness (lower bound, upper bound, and sufficient condition), and design a neural version of Subgraph-1-WL. To our knowledge, no existing subgraph-based literature shows this connection and the systematic analysis.
> 2. **Framework novelty**: No existing general framework that extends the use of subgraph aggregation to any GNN. Many existing methods (k-hop GNN, Ego-GNNs) can be seen as a special case of our framework. In that sense, our paper also unifies many existing rooted subgraph based methods.
> 3. **Design & Empirical novelty**: No existing designs propose as effective a concrete realization (all 3 types of encodings, D2C feature) as we did, which are essential for the superior real-world performance (SOTA on several datasets).
> 4. **Scalability novelty**: No existing subgraph-based GNN works address scalability issues involved in manifesting and message passing over the subgraphs, which are a *huge* issue for all subgraph-based methods in practice. A notable contribution of our paper is SubgraphDrop and random-walk based subgraph extractor, which greatly mitigates this issue without losing performance.

---

> > ### Author Response · Authors · 2021-11-18
> > **Response to architecture questions**
> >
> > >**Architecture:  their theoretical results do not seem to advocate for the necessity of these architectural choices (centroid encoding and context encoding)**
> > >**What is the actual reason for the arguably good empirical results? Is it due to the improved expressivity or due to engineering?**
> >
> > Firstly, our theory supports the usage of subgraph encoding and centroid encoding (Subgraph-1-WL also supports using centroid encoding, potentially making *HASH* over subgraphs more discriminative), but *does not* refute the usage of context encoding. In fact, using context encoding further improves the empirical expressiveness of GNN-AK, while theoretically analyzing its expressiveness is non-trivial in our Subgraph-1-WL paradigm, and thus we leave it to future work. As in most deep learning work, practical design often precedes theoretical understanding: likewise, GNN-AK is more empirically expressive *with* than  *without* context encoding, where the latter is fully supported by our theory through Subgraph-1-WL. To fully address the reviewer’s concern we evaluate GNN-AK without context encoding on all simulation datasets and two real-world datasets in appendix A.10 Table 8, where we observe that GNN-AK outperforms GNN-AK without context encoding, while the latter still significantly improves over base GNN. We also note that theory shouldn’t block designing better models: MLP is technically a universal approximator, but new models like CNN and transformer are designed to achieve better practical performance. Moreover, expressiveness is not always correlated with real-world performance (in fact, 1-WL already can distinguish many graphs in most  datasets, see Figure 3 of https://arxiv.org/pdf/2105.05911.pdf), while the trade-off between generalization and expressiveness plays an important role and is highly affected by designs.
> >
> > To help understand the actual reason for the superior performance of GNN-AK, Appendix A.10 Table 8 also shows additional ablation studies to decouple all designs and identify the root of the improvements. As we can see, the theoretically substantiated designs (GIN-AK w/o Ctx&D2C) indeed provide the largest improvement (42.3% error reduction on ZINC-12K, 7.69% absolute acc improvement on CIFAR10) over the base model, while context encoding and D2C provides additional benefit (additional  8.5% error reduction on ZINC-12K, additional 4.65% absolute acc improvement on CIFAR10). As we noticed that GIN-AK w/o Ctx&D2C offers comparatively less improvements on simulation datasets, we hypothesize that this is because of weak expressiveness of 1-layer base GIN when D2C is not used, and increasing base model’s depth is important. Hence we further studied the impact of the base model’s depth under the theory of Subgraph 1-WL in Table 9, and showed that with a deeper base model, GIN-AK w/o Ctx&D2C still provides a significantly large improvement over the base model. All observations in Table 9 are fully supported by proposition 1, we suggest the reviewer to read the **extensively updated Appendix A.10**.
> >
> > To summarize, our theory justified the significant performance improvement of GIN-AK without context encoding by Subgraph-1-WL, and the performance of GIN-AK without both D2C and context encoding by Subgraph-1-WL^*. Proposition 1 justified the empirical study of base model’s depth with and without D2C. Context encoding and D2C are important designs to benefit GNN-AK even further, which support the novelty and importance of our concrete realization. In future work we plan to theoretically prove the expressiveness benefit of context encoding and D2C within GNN-AK.
> >
> > >**Architecture: some of the choices of the authors (random-walks to extract the subgraphs, dropout on the ego-nets) introduce randomness in the algorithm which (1) sacrifices permutation equivariance (which is not necessarily bad), but (2) improves expressivity**
> >
> > SubgraphDrop **does not** introduce any randomness that sacrifices permutation equivariance, and it outputs **deterministic** embedding for an input graph G: if you pass graph G to GNN-AK-S several times during inference, the output won’t change from time to time. SubgraphDrop is similar to Dropout, which is not random during inference. Random-walk subgraph extractor does have the issue mentioned by the reviewer, however, we only use it for the PATTERN dataset as it is too dense for k-egonet to work.

---

> > > ### Author Response · Authors · 2021-11-18
> > > **Response to theory questions**
> > >
> > > >**Theory: The theoretical results up to Corollary 3.1 are similar to previous work … further arguments are required (e.g., substructure counting, or graph property prediction - possibility/impossibility results).**
> > >
> > > We thank the reviewer’s suggestion and would like to study the suggested theoretical problem in future. Nevertheless, we argue that the contributions (theory, realization designs, scalability) of the paper already makes the paper too crowded in terms of pages limitation. Moreover, our work also has significant empirical impact (as discussed earlier in our response) and is not primarily motivated to be theoretical in nature.
> > >
> > > >**Theory: I don’t understand the following sentence: “This opens a future direction of generalizing rooted subgraph to general subgraph (as in k-WL) while keeping number of subgraphs in $O(|V|)$’’.**
> > >
> > > GNN-AK uses rooted subgraphs. In terms of scalability, rooted subgraphs are scalable as there are only $|V|$ rooted subgraphs. However the rooted subgraph pattern is also limited, as shown in Theorem 4 that it is not possible to capture the difference between a hard pair of non-isomorphic graphs even with the relatively large 4-egonets. On the contrary, $k$-WL can be more expressive than GNN-AK but it uses $|V|^k$ number of subgraphs, which massively hurts its scalability. Thus, it opens an interesting question about how to design or choose “few, but good” subgraphs for message passing -- intuitively, can we get an outsized portion of expressivity gains (in theory or practice) by using $O(|V|)$ subgraphs rather than directly using $|V|^k$?  Doing so would suggest better expressivity and scalability tradeoffs.
> > >
> > > >**Proof: Theorem 4: (1) Notation clash … and (2) there are clarity issues in the proof. … Does this proof imply that with $\ge 5$-egonet  you might be able to distinguish two CFI graphs?**
> > >
> > > We have significantly updated the proof of Theorem 4 in Appendix A.5 in detail. At the end of the proof, we also answer the reviewer’s question about $\ge 5$-egonet. The short answer is it depends on the base graph of CFI construction, and we provide a sketch of why even with $\ge 5$-egonet, under certain conditions our method may still fail to distinguish two CFI graphs.
> > >
> > > >**Theory: More clarity is needed in Conjecture 1 with respect to the magnitude of $k$.**
> > > >**Proof: As far as I understood, this is based on a conjecture rather than a proven fact.**
> > >
> > > We change Conjecture 1 to Proposition 1, and give a formal proof in Appendix A.6. The proposition is about a sufficient condition for successfully distinguishing two non-isomorphic graphs, regardless of how the subgraph is defined (more specifically, $k$-egonet with any $k$)). In other words, if the conditions in Proposition 1 are not satisfied for a pair of graphs, then GNN-AK will fail in distinguishing them.
> > >
> > > >**Theory: Can you theoretically compare ego-net enabled GNNs with other modern GNNs**
> > >
> > > As we mentioned, several works can be viewed as a special weaker realization of our general framework. For other works like [Bodnar et al. NeurIPS 21] and [Thiede et al. NeurIPS 21], they share a similar expressiveness argument as us: strictly more powerful than 1-WL and no less powerful than 3-WL. However their preprocessing requires solving subgraph isomorphism tests which is considerably heavier than our linear preprocessing.

---

> > > > ### Author Response · Authors · 2021-11-18
> > > > **Last**
> > > >
> > > > >**Experiments: What version of GNN-AK do you use in Tables 1 and 2?**
> > > >
> > > > We use GNN-AK with all the proposed concrete realizations: 3 types of encodings with D2C. However we have done extensive additional ablation study in Appendix A.10 Table 8 and Table 9 based on your request.
> > > >
> > > > >**Experiments:  I suspect that this method might suffer from scalability issues when dealing with denser graphs.**
> > > >
> > > > Indeed, scalable designs are required when handling such dense graphs, and this is exactly why we attend to this component in our paper. Random-walk subgraph extraction is designed for this case, which we apply to the dense PATTERN dataset (generated from SBM, see dataset statistics in Appendix A.8 ), to circumvent the $O(n^2)$ complexity. What’s more, SubgraphDrop is designed to further reduce memory and runtime overhead without sacrificing performance.
> > > >
> > > > --------------------
> > > > We thank the reviewer once more for their detailed feedback -- it has inspired extensive additional updates to our work and has made it considerably stronger. In light of these updates, we kindly ask the reviewer to consider raising their score if their concerns were appropriately addressed.

---

> > > > > ### Comment · Reviewer_MkR2 · 2021-11-22
> > > > > **Comments after rebuttal**
> > > > >
> > > > > I would like to second the new comments made by Reviewer oHnh and add some thoughts to the discussion. In particular:
> > > > > -	**Contributions and design choices**: It is totally fine and perhaps desirable to aim for higher practical performance when experimenting. However, it should always be clear what are the novel contributions and what are the implementation details. Currently, the difference is a bit shady. From my point of view, the novel contributions are up to, and including section 3, i.e., proposing ego-nets as the main GNN building block, together with the theoretical analysis. Hence, in terms of presentation, I would expect an experimental evaluation with a vanilla GNN-AK that closely follows the theoretical results and then (optionally) a more involved version, aiming for SOTA.
> > > > > -	**Prior work on ego-net based GNNs**: I still find the comparison with You et al., AAAI’21 insufficient (the authors probably forgot to cite it in the updated version). The main idea in both papers is essentially the same and the differences the authors mentioned mainly have to do with the implementation. I acknowledge that the present work goes deeper and generalises ID-GNNs, but the authors should have been more upfront in that respect since currently the contributions might be overstated.
> > > > > -	For the authors’ reference, recently I found out a concurrent work (“Nested Graph Neural Networks”, Zhang and Li, NeurIPS’21) that proposes the same idea, except for the concrete realization. Obviously, the authors could not know this, but I advise referencing it in an updated version (note that this paper contains strong complementary theoretical results).
> > > > > - **Minor**:
> > > > >     - I would be also interested in knowing more about the splits used for the TU dataset evaluation.
> > > > >     - Did the authors use the number of parameters (100k or 500k) the ZINC benchmark requires?
> > > > > - **Overall**: on the positive side, the paper has improved significantly w.r.t. its initial version. However, ego-net based GNNs have been independently proposed several times (not only concurrently with this work), hence I believe it would be fairer to present the paper as a way of revisiting them and exploring them more in-depth (theoretically and implementation-wise, scalability including), rather than as an entirely new contribution to the field (e.g., the authors mentioned that they are the first to propose Subgraph-1-WL, but this is not really true; the method has been reinvented, it just goes by different names). It is true that several aspects of the approach appear for the first time in this paper, but the current presentation of the contributions is misleading and makes me sceptical about recommending publication.
> > > > >
> > > > > **Suggestions that might strengthen the theoretical contributions** (not affecting my decision, but the authors might find them useful):
> > > > >    - Given that the context encodings seem to provide a boost in performance – see the subgraph counting tasks, the reason for that is unclear to me - perhaps the authors might be interested in including this in their theoretical analysis, which would be also a novel contribution.
> > > > >    - Although I agree that scalability is a major advantage of ego-nets, a theoretical comparison with other modern GNNs would be very insightful and convincing.  I am making this suggestion because currently there is an explosion of papers with increased GNN expressivity, but they rarely compare with each other theoretically, making it hard for the community to understand what to adopt and where we should be heading towards.

---

> > > > > > ### Author Response · Authors · 2021-11-25
> > > > > > **Response to Reviewer MkR2 (Second Round)**
> > > > > >
> > > > > > >**In terms of presentation, I would expect an experimental evaluation with a vanilla GNN-AK that closely follows the theoretical results and then (optionally) a more involved version, aiming for SOTA.**
> > > > > >
> > > > > > We have revised the paper based on your suggestions. Please have a look at Section 4, marked in blue. Specifically, we rename the proposed model with auxiliary components (context encoding and D2C) as GNN-AK+, and the one that strictly follows Subgraph 1-WL theory as GNN-AK. Also we have included both GNN-AK and GNN-AK+ in the experiment section, with ample additional results in the Appendix given space constraints.
> > > > > >
> > > > > > >**I still find the comparison with You et al., AAAI’21 insufficient. The main idea in both papers is essentially the same...**
> > > > > >
> > > > > > We apologize for mistakenly forgetting to mention the paper in the first revision, and we have included it in Section 2 now.
> > > > > >
> > > > > > Nevertheless, we feel the main idea in both papers is **fundamentally different**, but the two papers share some similarity in terms of architecture and implementation. The central idea of ID-GNN is to label each node with an *identity feature* while still keeping the model permutation invariant and inductive (notice that directly adding node index indicators results in a transductive and permutation sensitive model). The authors designed a clever way to achieve both goals: for a node $v$ it uses $K$-egonet of $v$ where K is the depth of a normal GNN so that the $K$-egonet of $v$ is the *receptive field* of $v$ in normal $K$-depth GNN, which is “identical to the widely used mini-batch version of GNNs (Hamilton, Ying, and Leskovec 2017; Ying et al. 2018)”. Then, the authors mark the root of the $K$-egonet differently from all other nodes in message passing, which achieves adding identity features to each node $v$ while still maintaining inductiveness and permutation invariance. To summarize, the $K$-egonet is used to help inject the identity feature to each node, the $K$-egonet is exactly the *receptive field* of $v$ in normal $K$-depth GNN, and exactly $K$ number of graph convolutions is applied to align with the normal $K$-depth GNN: the resulting model ID-GNN is exactly the same as mini-batch version GNN after removing the injected identity feature (this is clearly stated in Proposition 1 in the paper).
> > > > > >
> > > > > >
> > > > > > GNN-AK is instead motivated by representing each node with the encoding of its rooted subgraph, where $k$-egonet is just a special case of rooted subgraph and $k$ can be any value. GNN-AK has multiple (outer) layers as Subgraph-1-WL needs multiple iterations to converge to a stable coloring (equitable partition). Within each (outer) layer, GNN-AK applies a general base GNN (in our experiments we use 1-layer GNN) to each rooted subgraph, and use the *graph-level representation* of the rooted subgraph as a new presentation for the root node of the subgraph. This demonstrates that these two models are fundamentally different, and the similarity is mainly in the implementation choice of using ego-net.
> > > > > >
> > > > > >
> > > > > > >**For the authors’ reference, recently I found a concurrent work (“Nested Graph Neural Networks”, Zhang and Li, NeurIPS’21) that proposes the same idea...**
> > > > > >
> > > > > > We have checked the paper and this paper does share similarity with our paper, however it only has 1 outer layer, and applies GNN independently to all subgraphs, which is very similar to the G-meta (Huang & Zitnik, 2020) approach (also without any solution for scalability issues). The paper was made available on arXiv on **Oct 25 2021** which is 20 days after the ICLR submission deadline. Thus, we consider this work concurrent. We will cite the paper in the final version of the paper.
> > > > > >
> > > > > > >**I would be also interested in knowing more about the splits used for the TU dataset evaluation.**
> > > > > >
> > > > > > Previously we were using a random split, but we are working on the split mentioned by the reviewer and will present the result as soon as possible after we finish the experiments.
> > > > > >
> > > > > > >**Did the authors use the number of parameters (100k or 500k) the ZINC benchmark requires?**
> > > > > >
> > > > > > The exact number of parameters of the presented model with 0.080 MAE is 1000K. We have rerun a model with 500K budget and its MAE is 0.086.
> > > > > >
> > > > > > >**The authors mentioned that they are the first to propose Subgraph-1-WL, but this is not really true; the method has been reinvented, it just goes by different names**
> > > > > >
> > > > > > We have acknowledged in paper that using subgraph is not a new idea and summarized significant related work in Section 2. But, to the best of our knowledge, we have never seen a paper presenting a specific notion of WL with a rooted subgraph as we have done — it would be helpful if the reviewer could reference the specific papers they have in mind here.  Also, the analysis under Subgraph-1-WL is systematic and comprehensive: we give lower bounds of its expressiveness, novel upper bound of its expressiveness by Theorem 4, and sufficient conditions which have never appeared before in prior literature to our knowledge.

---

> > > > > > > ### Author Response · Authors · 2021-11-26
> > > > > > > **Updated Results on TUDataset**
> > > > > > >
> > > > > > > We have got the updated result for TUDataset, we use the same split, same configuration and same evaluation as CIN. Please see the following table:
> > > > > > >
> > > > > > > | Method   | MUTAG | PTC| PROTEINS | NCI1|  IMDB-B| RDT-B |
> > > > > > > | ---- | ------ |------ |----- |------ |------ |----|
> > > > > > > |CIN|92.7 ± 6.1 |68.2 ± 5.6 |77.0 ± 4.3 |83.6 ± 1.4| 75.6 ± 3.7| 92.4 ± 2.1|
> > > > > > > |GIN-AK|91.3 ± 7.0|67.8 ± 8.8|77.1 ± 5.7|85.0 ± 2.0|75.0 ± 4.2|94.8 ± 0.8|
> > > > > > >
> > > > > > > The performance of GIN-AK over IMDB-B is not improved because each graph in the dataset is an egonet, and hence all nodes have the same rooted subgraph -- the whole graph.
> > > > > > >
> > > > > > > The performance of MUTAG and PTC is very unstable, given these datasets are too small: 188 and 344, respectively, and the evaluation method is based on average 10 validation *curves* over 10 folds.

---

> > > > > > > > ### Comment · Reviewer_MkR2 · 2021-11-27
> > > > > > > > **Recommendation**
> > > > > > > >
> > > > > > > > The decision on this paper has been a hard one. Mainly the fact that (1) there is a large body of related work studying the idea of ego-net based GNNs (some concurrent, others not) and (2) that this paper's contributions were unclearly presented and positioned in the context of the previous work (e.g., see the discussion about the design choices of the concrete realisation), have placed the paper on the borderline.
> > > > > > > >
> > > > > > > > That said, although still voicing my concerns, I will stand on the positive side and vote for weak acceptance, mainly due to the fact that I have honestly appreciated the hard work the authors have done to improve their paper (especially in such a small time window) and address the reviewers concerns (successfully in several cases). I urge the authors to be clear about all the related work and their contributions in the next/final version of their paper and consider the other suggestions that have been made by the reviewers.
> > > > > > > >
> > > > > > > > Final minor comments:
> > > > > > > > - Proposition 2 should have been phrased as "at least as powerful", not "more powerful" (except if you find a counterexample).
> > > > > > > > - Ideally, please show the results with the plain GNN-AK in all the tables. Also please use the results of either 100k or 500k params on the ZINC dataset for consistency with the leaderboards and fill in the missing results in table 10 (and change the caption).

---

> > > > > > > > > ### Author Response · Authors · 2021-11-28
> > > > > > > > > **Thank you**
> > > > > > > > >
> > > > > > > > > We thank the reviewer's suggestive feedback and the recognition for the additional work we did. Based on the reviewer's final thought, in the final paper we will
> > > > > > > > >
> > > > > > > > > 1) clearly state the difference between our work and previous work;
> > > > > > > > >
> > > > > > > > > 2) summarize all concurrent works presented in NeurIPS 2021 and ICLR 2022 submissions (there are 2 papers being similar to ours, mentioned by Reviewer oHnh) ;
> > > > > > > > >
> > > > > > > > > 3) revise Proposition 2 accordingly, as the reviewer's comment is more precise;
> > > > > > > > >
> > > > > > > > > 4) include plain GNN-AK's result in all the tables;
> > > > > > > > >
> > > > > > > > > 5) use 500k-params constraint on the ZINC dataset.
> > > > > > > > >
> > > > > > > > > We will also open source our code soon. Thank you again for your time on the review.

---

### Author Response · Authors · 2021-11-19
**Summary of the major revision**

We are grateful to all reviewers’ constructive feedback and questions; they have given our paper direction for a significantly stronger revision. We summarize all changes that we have made below.  All changes are marked in blue in the updated submission.
1. We added several important references to Section 2, and discuss the novelty of GNN-AK compared to these methods.
2. We changed Conjecture 1 to Proposition 1, and proved it formally in Appendix.A.6.
3. We updated the proof of Theorem 3 to be more formal, in Appendix.A.4.
4. We updated the proof of Theorem 4 to be more detailed and formal, in Appendix.A.5.
5. We provide dataset statistics in Appendix.A.8.
6. We added the stronger  CIN[1] baseline to main Table 3. Also, we conducted new experiments on 6 datasets from TUDataset in Appendix.A.11, where GNN-AK outperforms CIN and all other baselines significantly.
7. We conducted an extensive ablation study on Appendix.A.10 on all simulation datasets and two real-word datasets, by decoupling all components of GNN-AK. The results help on identifying the source of performance improvement: summarily, the major improvement derives from practical instantiations related to theoretical ideas from Subgraph-1-WL, while other practical designs (Context encoding and D2C) consistently give more marginal performance improvements.


**Reference**:

[1]Bodnar, Cristian, et al. 2021 "Weisfeiler and lehman go cellular: Cw networks."

---

### Decision · Program_Chairs · 2022-01-22

**Decision:**

Accept (Poster)

**Comment:**

In standard message-passing GNNs (MPNNs), one step at any node u involves receiving state/embedding information from all of u’s neighbors, and then updating u’s state as a function of these messages and of u’s own current state. Thus, the communication pattern at every step is that of a star topology (a graph with u at its “center”, and with u connected to all its neighbors, and with no other edges). However, it is well-known that the expressive power here is bounded by that of the 1st order Weisfeiler-Leman isomorphism test (1-WL). This paper then takes the natural step of generalizing the star topology to more general ones (e.g., k-hop egonets, the subgraph induced by the nodes of distance at most k from u). It is shown that this framework is strictly more powerful than 1-WL and 2-WL (however, as pointed out by a referee, this is actually a weaker version of 2-WL that is equivalent to 1-WL), and is at least as powerful as 3-WL. Subgraph-sampling approaches that improve efficiency are also introduced. It is shown that this method beats the SOTA for some number of well-known graph-ML problems.

It looks like this paper has a very strong overlap with the NeurIPS 2021 paper "Nested Graph Neural Networks" (https://openreview.net/forum?id=7_eLEvFjCi3). Both papers use rooted subgraphs (k-hop ego-nets) to replace the k-hop rooted subtrees in traditional GNNs, and both use a base GNN over the rooted subgraph to compute a subgraph representation as the node representation while pooling the node representations into a graph representation. Both papers claim to outperform 1-WL in expressive power; both use distance to center node in order to enhance subgraph node features. The authors are urged to compare and contrast the two papers in the camera-ready version.

---

> ### Public Comment · ~Lingxiao_Zhao1 · 2022-01-30
> **Response to PC's Concern: Similarity to Nested GNN**
>
> The Nested GNN paper does share a very similar idea with GNN-AK. In fact, it did shock us when we first saw the paper. However we already had our first version code (without sampling part and theoretical analysis) available on April 23, 2021 in the first author's private Github repository (we can public it for verification), which is even significantly earlier than NeurIPS 2021 submission deadline.
>
> On the other hand, our paper is still very different from the Nested GNN paper from several perspectives:
> 1. We proposed Subgraph-1-WL and analyzed it theoretically, which motivated the design of GNN-AK. Our theoretical result is different from Nested GNN.
> 2. As motivated from Subgraph-1-WL, GNN-AK has many outer layers (in fact we always keep base model's #layer as 1), which is the same as the iterations of WL algorithm. However the Nested GNN only has 1 outer layer, and their structure is the same as G-Meta (https://proceedings.neurips.cc/paper/2020/file/412604be30f701b1b1e3124c252065e6-Paper.pdf).
> 3. Motivated from EgoGNN paper, we designed context encoding to fully explore rich information inside all intermediate embeddings. Nested GNN doesn't have the same design.
> 4. SubgraphDrop is designed to solve the scalability issue, and Nested GNN doesn't address this important issue.
>
> We will further emphasize the difference between GNN-AK and Nested GNN in our camera-ready submission.